# Identification of conserved frontal neurophysiological markers of cognitive flexibility in humans and rats
Andre Der-Avakian[1,9], Samuel A. Barnes[1,9], Ty Lees [2,9], Hans S. Schroder[2], Brian D. Kangas [2], Samantha R. Linton [2], Stefanie Nickels [2], Mykel A. Robble[2], Micah Breiger[2], Ann M. Iturra-Mena[2], Rachel Lobien[2], Sarah Perlo[2], Emilia F. Cárdenas[2], Genevieve P. Nowicki[2], Zeyun Wu[3], Hongyi Pan[3], Daniel G. Dillon [2], James P. Kesby [4,5], Jack Bergman[2], William A. Carlezon Jr[2], Victoria B. Risbrough[1,6], Eran Mukamel [3], Stefan Leutgeb [7] & Diego A. Pizzagalli [2,8] ✉

Cognitive flexibility broadly describes behavioral alterations made in response to environmental changes and is fundamental for survival. While human and non-human animal assessments of cognitive flexibility are available, a systematic cross-species comparison of behavioral, neurophysiological, and computational markers of cognitive flexibility has not been reported. Using versions of a probabilistic reversal learning task aligned between humans and rats, electroencephalogram recordings reveal a frontal reward positivity (RewP) associated with unexpected reward outcomes. Reinforcement Q-learning models of both species' task behavior reveal that prediction error (PE) magnitude was significantly related to RewP amplitude. The stimulant drug modafinil alters PEs in rats without affecting the RewP in either species. These findings reveal analogous neurophysiological markers associated with PEs in humans and rats using equivalent tasks and identical computational analyses. This translational approach may improve the predictive validity of tests for novel pharmacotherapies and accelerate neuropsychiatric treatment by assessing neural mechanisms conserved across species.

Cognitive flexibility broadly refers to the ability to alter behavior in response to a changing environment[1–3], and such adjustments are a vital component of navigating everyday life[4,5]. It describes the balance between repeating behaviors that yield beneficial outcomes toward a particular goal and modifying behaviors in response to environmental changes[3,6]. This modification of behavioral strategy is key to cognitive flexibility and enables previously learned rules to be updated to maximize rewarded outcomes. Cognitive flexibility has been studied across several key domains, including set-shifting, working memory, and reversal learning[6]. Moreover, human research has found that cognitive flexibility is associated with resilience to stress and negative life events[4,7] and overall higher quality of life[8]. Conversely, deficits in cognitive flexibility have been documented in a multitude of psychiatric disorders, including mood disorders[9–11], schizophrenia[12,13], obsessive

compulsive disorder[14,15], and substance use disorder[16]. If the neurobiological processes underlying cognitive flexibility were better understood, they could be targeted for treatment in these patient populations.

Reversal learning tasks, particularly probabilistic versions, are ideal for assessing cognitive flexibility[17–19]. Although several variations of the probabilistic reversal learning (PRL) task exist, they are all based on a fundamental reinforcement learning paradigm whereby provided feedback is not always informative and the stimulus associated with the higher probability of reward is subject to change. For example, in a common PRL procedure, subjects are presented with two distinctive stimuli, one of which is associated with a high (e.g., 80%) probability of reinforcement, while the other is associated with a low (e.g., 20%) probability of reinforcement. By sampling both stimuli, subjects learn to maximize responding to the stimulus

---

[1]Department of Psychiatry, University of California San Diego, La Jolla, CA, USA. [2]Department of Psychiatry, McLean Hospital & Harvard Medical School, Belmont, MA, USA. [3]Department of Cognitive Science, University of California San Diego, La Jolla, CA, USA. [4]Queensland Centre for Mental Health Research, Wacol, QLD, Australia. [5]Queensland Brain Institute, The University of Queensland, St. Lucia, QLD, Australia. [6]Veterans Affairs Center of Excellence for Stress and Mental Health, La Jolla, CA, USA. [7]Neurobiology Section and Center for Neural Circuits and Behavior, University of California San Diego, La Jolla, CA, USA. [8]Noel Drury, M.D. Institute for Translational Depression Discoveries, University of California, Irvine, CA, USA. [9]These authors contributed equally: Andre Der-Avakian, Samuel A. Barnes, Ty Lees. ✉e-mail: dpizzaga@hs.uci.edu

associated with a greater probability of reinforcement (i.e., target stimulus). After a pre-determined criterion (e.g., consecutive responses for the target stimulus), the reinforcement contingencies are reversed, and subjects need to identify the new target stimulus (identified as a reversal). Thus, PRL tasks require subjects to ignore occasional negative outcomes while responding to the target stimulus and avoid perseverating on a previously successful action when the reward contingencies are reversed. Individuals with major depressive disorder are sensitive to negative events, and for example, may respond to the occasional misleading negative feedback following a target response by prematurely responding for the other (i.e., non-target) stimulus[20], whereas patients with obsessive compulsive disorder may perseverate on a stimulus that is no longer the target stimulus, despite the new relatively low reinforcement probability[21]. In either case, patients may complete fewer reversals, indicative of impairment in adaptive behavior.

Expression of flexible behavior during a PRL task requires successful and rapid reinforcement learning[22], in particular, the ability to parse the likelihood of reward and identify the target stimulus. As a result, a violation of the expected outcome, such as the omission of reward following a target response (i.e., due to probabilistic feedback), should elicit a reward prediction error (PE). Frequent, but not occasional, PEs should facilitate a change in response strategy. PEs have been associated with changes in event-related potentials (ERPs) over frontocentral electrode sites, namely the Reward positivity (RewP)[23,24]. Originally named the feedback-related negativity[25,26], and identified as a negative deflection following unexpected negative feedback (i.e., a negative PE) relative to expected positive feedback[27,28], more recent evidence clarified that the RewP is driven by the response to reward feedback rather than non-rewarded outcomes[23,29,30]. Evidence suggests that the FRN/RewP originates in the anterior cingulate cortex (ACC)[31–33]. Alterations in the activity of midbrain dopamine neurons are known to code PEs[34–36], whereby negative PEs elicit decreased firing of these neurons and phasic activation encodes positive PEs. This dopaminergic signal projects to the striatum and ACC and may modulate the FRN/RewP, which can subsequently predict behavioral adjustments[26,37,38].

Non-human animal versions of the PRL task are critical for advancing our understanding of cognitive flexibility. In particular, the operant nature of the task enables translation across species; moreover, task behavior is well-characterized by reinforcement learning models, enabling application of the same (or very similar) models to data across species[39]. To study this phenomenon in non-human animals, rodent versions of the PRL task have been developed[40,41]. While human and non-human versions of the task are conceptually similar and may yield comparable behavioral results, to our knowledge, a systematic cross-species comparison of behavioral, computational, and neurophysiological markers of cognitive flexibility has not been reported. Such a comparison would be useful for drug development because putative treatments that demonstrate therapeutic behavioral effects and target engagement in non-human animals could be used in parallel human testing to accelerate drug discovery.

Thus, we modified human and rodent versions of a PRL task to align several parameters, enabling the comparison of behavioral performance across species. We measured the electroencephalogram (EEG) during the task to determine whether unexpected rewards were associated with a frontal RewP. Additionally, we applied a reinforcement Q-learning model to model behavior and linked it to EEG data from both species to compare the relationship between action values and frontal neurophysiological signals. Lastly, we administered comparable doses of modafinil, an indirect dopamine (DA) agonist that also has norepinephrinergic effects, to both species to determine whether modulation of dopaminergic signaling similarly altered the behavioral and neurophysiological indices of cognitive flexibility in humans and rats. Prior studies have shown that DA agonists increase reward-related ERPs during reward learning[42] and modafinil increases DA signaling in the striatum (via the inhibition of DA transporters), the neural substrate for error prediction[43]. Critically, we identified behavioral, computational, and neurophysiological markers of cognitive flexibility that were similar across species, providing a robust platform that could hasten treatment development.

## Results

We instructed humans ($n = 54$) and trained rats ($n = 11$) to perform functionally identical versions of a PRL task (Supplementary Fig. 1) that consisted of 300 trials and reinforced responses probabilistically using an 80%/20% schedule (see "Methods" section). For both species, the reward contingencies reversed after eight consecutive correct responses, and EEG recordings were obtained during task performance (see "Methods" section).

### Behavioral indices of cognitive flexibility and reinforcement metrics are consistent across humans and rats

During a single test session, humans completed $4.4 \pm 0.23$ (mean ± SEM) reversals per 100 trials (Fig. 1A), whereas rats completed $2.6 \pm 0.40$ (Fig. 1B). Repeating a target response after reward delivery (i.e., target win-stay) and abandoning the target response after non-reward delivery (i.e., target lose-shift) reflect responsiveness to positive and negative feedback, respectively. In both humans (Fig. 1A) and rats (Fig. 1B), target win-stay probability was significantly greater than target lose-shift probability [humans: $t(53) = 13.93$, $p < 0.001$; rats: $t(10) = 6.20$, $p < 0.001$]. Increased target win-stay and reduced target lose-shift responding facilitated more reversals, as reflected by significant positive correlations between target win-stay probability and reversals in humans (Pearson $r(52) = 0.44$, $p < 0.001$; Fig. 1C) and rats ($r(9) = 0.79$, $p = 0.004$; Fig. 1D), and significant negative correlations between target lose-shift probability and reversals in humans ($r(52) = -0.58$, $p < 0.001$; Fig. 1C) and rats ($r(9) = -0.73$, $p = 0.012$; Fig. 1D). Thus, in both species, greater sustained responding for reinforced target responses, despite occasional misleading feedback, was associated with better task performance.

### Reinforcement learning models identify consistent patterns of behavioral responding across species

To gain a deeper insight into the behavioral mechanisms underlying PRL performance, we fit several Q-learning models to behavior (see Supplementary Methods). Consistent with our previous work[41,44], we found that the model consisting of three free parameters (learning rate, inverse temperature, and forget parameters) best fit PRL performance (Supplementary Table 1). Moreover, parameter recovery and posterior predictive checks (Supplementary Fig. 2) demonstrated recoverable parameter estimates and alignment between simulated and observed PRL performance.

As subjects perform the PRL task, the value of each chosen action (i.e., Q-value) is updated based on reinforcement. Thus, during the task, the value of the target stimulus fluctuates to reflect the changing probability of reward delivery (see Supplementary Fig. 3A, B for representative Q-values across a test session for both species). As expected, humans assigned greater Q-values to target stimuli ($0.710 \pm 0.007$) than non-target stimuli ($0.329 \pm 0.01$) [$t(106) = 25.51$; $p < 0.001$] (Supplementary Fig. 3C), as did the rats ($0.550 \pm 0.036$ vs. $0.275 \pm 0.034$) [$t(20) = 6.08$, $p < 0.001$] (Supplementary Fig. 3C), indicating that both species learned to appropriately value stimuli based on experience throughout the task.

The beta parameter reflects the degree to which subjects explore both actions (lower beta value) vs. exploit the highest value action (higher beta value)[45]. Higher beta values were significantly associated with more reversals in humans ($r(52) = 0.65$, $p < 0.001$; Fig. 1E) and rats ($r(9) = 0.69$, $p = 0.019$; Fig. 1F). Thus, consistent with the target win-stay and lose-shift correlations described above, exploiting high value actions and limiting exploration to periods when those actions become less favorable (i.e., after reversals) was an adaptive strategy for both species. Conversely, there was no association between alpha or forget parameters and reversals in either species (Supplementary Fig. 4).

### Electrophysiological markers associated with reward expectancy are consistent across humans and rats

As humans and rats performed the PRL task, continuous EEG was recorded and averaged across rewarded and non-rewarded trials. A RewP emerged in frontal recording sites (e.g., frontocentral electrode (FCz) in humans (Fig. 2A); ACC in rats (Fig. 2B)) in response to positive vs. negative feedback. Highlighting spatial specificity, these effects were not present at parietal

**Fig. 1 | PRL behavior was comparable between humans (left) and rats (right) performing similar versions of the task.** Although humans (*n* = *54*; **A**) generally completed more reversals than rats (*n* = 11; **B**), the probability of repeating a previously rewarded target response (win-stay) was greater than the probability of abandoning a previously unrewarded target response (lose-shift) in both species. Target win-stay probability was positively associated with the number of completed reversals in both humans (**C**) and rats (**D**), while target lose-shift probability was negatively associated with reversals in both species. Greater beta values, indicating a greater likelihood to exploit known information to maximize rewards, positively correlated with the number of completed reversals in humans (**E**) and rats (**F**). Reversal data are presented using box plots marking the median value (center line), the 25th and 75th percentiles (the outer edges), and ± 1.5 times the interquartile range (the whiskers), as well as the mean (triangle) ± standard error of the mean (error bars).

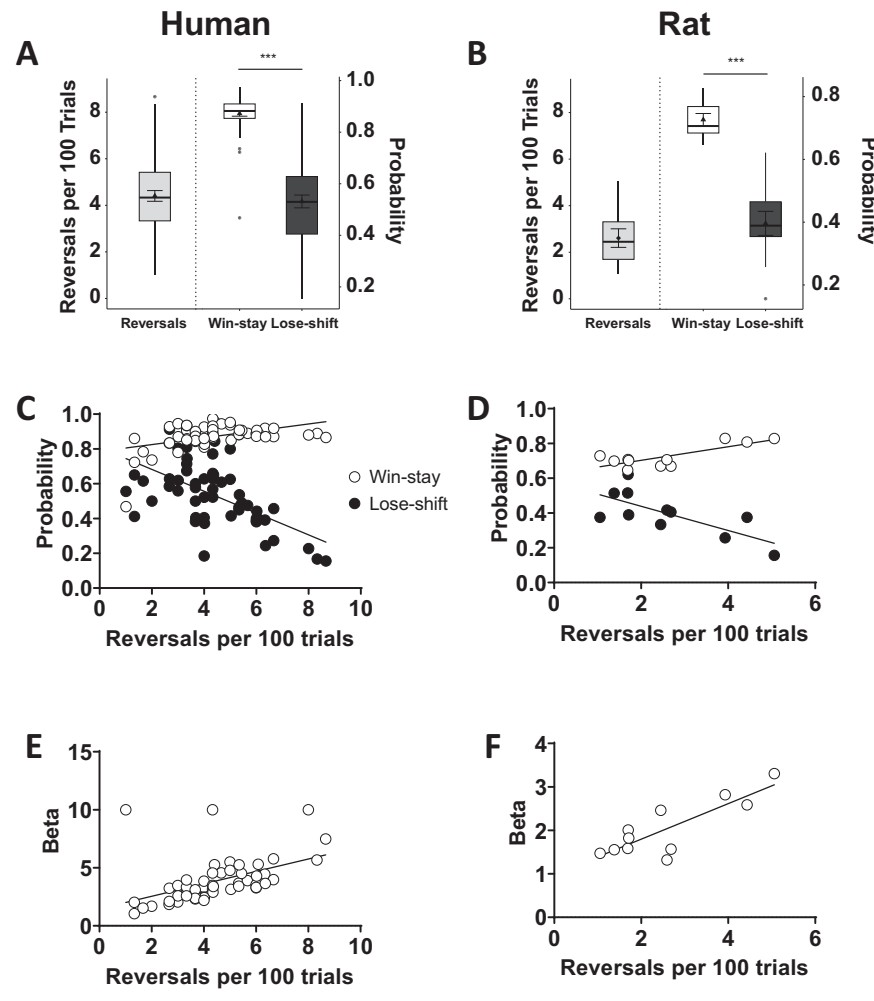

electrodes in either species (*p's* > 0.168). The RewP peaked in the human FCz electrode approximately 200 ms following feedback and approximately 100 ms after reward feedback in the rat ACC, and reflected a more positive frontal signal for a rewarded vs. non-rewarded response (consistent with a positive PE). When broken down further into the four possible trial types, the average RewP amplitude was greater for rewarded compared to unrewarded trials after both target and non-target responses in both humans (Fig. 2C) and rats (Fig. 2D). In humans, a 2-way ANOVA revealed a significant Response × Feedback interaction [$F(1,33) = 6.10$, $p = 0.019$], and Bonferroni post-hoc tests revealed significantly greater voltage during rewarded vs. unrewarded target trials ($p = 0.002$). Moreover, a 1-way ANOVA revealed a significant linear effect of trial type [$F(3,99) = 30.23$, $p < 0.001$], and Bonferroni post-hoc tests revealed progressively increasing voltage that scaled with expected PE magnitude: Target/No Reward (expected to elicit the largest negative PE), Non-target/No Reward, Target/Reward, Non-target/Reward (expected to elicit the largest positive PE). In rats, a 2-way ANOVA did not reveal a significant Response × Feedback interaction, but did reveal a significant main effect of Feedback [$F(1,9) = 37.73$; $p < 0.001$], and Bonferroni post-hoc tests revealed significantly more positive voltage during rewarded vs. unrewarded target trials ($p < 0.001$). Critically, like humans, a 1-way ANOVA in rodents revealed a significant linear effect of trial type [$F(3,27) = 18.52$, $p < 0.001$], and Bonferroni post-hoc tests revealed progressively increasing voltage in the same order as humans.

**Prediction errors associated with reward expectancy are consistent across humans and rats**
Based on the value assigned to the chosen option on any trial, a PE can be computed depending on the outcome of that trial. For example, rewarded target responses (i.e., informative feedback) should elicit small positive PEs, whereas non-rewarded target responses (i.e., misleading feedback) should elicit large negative PEs[46,47]. Indeed, the PEs for rewarded and unrewarded outcomes were positive and negative, respectively, and PE magnitude was greater when the outcome of the selected action was unexpected (Fig. 2E, F). In humans, a 2-way ANOVA revealed a significant main effect of Feedback ($F(1, 53) = 22,570.94$, $p < 0.001$), but no Response × Feedback interaction. Post-hoc tests revealed significantly greater PEs during rewarded vs. unrewarded target trials ($p < 0.001$). Consistent with the linear effect of the four trial types on RewP voltage, 1-way ANOVA revealed a significant linear effect of trial type [$F(3,159) = 3571.61$, $p < 0.001$], and Bonferroni post-hoc tests revealed progressively increasing PE values in the same order as described for RewP voltage. In rats, a 2-way ANOVA revealed a significant main effect of Feedback ($F(1, 10) = 5698.12$, $p < 0.001$), but no Response × Feedback interaction. Post-hoc tests revealed significantly greater PE during rewarded vs. unrewarded target trials ($p < 0.001$). Similar to humans, the 1-way ANOVA across all four trial types in rodents revealed a significant effect [$F(3,30) = 882.49$, $p < 0.001$], due to progressively increasing PEs. Interestingly, in both species, PEs were more negative for unrewarded target vs. non-target responses and more positive for rewarded non-target vs. target responses (all *p*'s < 0.001), suggesting that reward expectancy was greater following target vs. non-target responses in both groups.

**Relationships between behavioral and electrophysiological measures of PEs**
Next, we investigated associations between model parameters and electrophysiological markers. Using linear regression, we used trial-level data to

**Fig. 2 | Frontal feedback-locked event-related potentials (ERPs) and prediction error (PE) values were comparable between humans (left) and rats (right) during performance of the PRL task.** Frontal (FCz in humans, anterior cingulate cortex in rats) ERPs were more positive during rewarded vs. unrewarded target responses in both humans (**A**) and rats (**B**), indicative of the reward positivity (RewP). The pattern of voltage changes during the RewP period across all four trial types was similar between humans (*n* = 34; **C**) and rats (*n* = 11; **D**). A nearly identical pattern emerged with regard to PE values in both humans (*n* = 34; **E**) and rats (*n* = 11; **F**). ERP traces are presented for all trial types (rewarded target = blue solid line; non-rewarded target = red solid line; rewarded non-target = blue dotted line; non-rewarded non-target = red dotted line) and expectancy-based difference waveforms (Expected = Rewarded Target – Non-rewarded Non-Target, and Unexpected = Rewarded Non-Target – Non-rewarded Target) and were averaged across all subjects and triggered to the tone predicting the presentation or omission of reward. Voltage and PE data are presented using box plots marking the median value (center line), the 25th and 75th percentiles (the outer edges), and ± 1.5 times the interquartile range (the whiskers), as well as the mean (triangle) ± standard error of the mean (error bars).

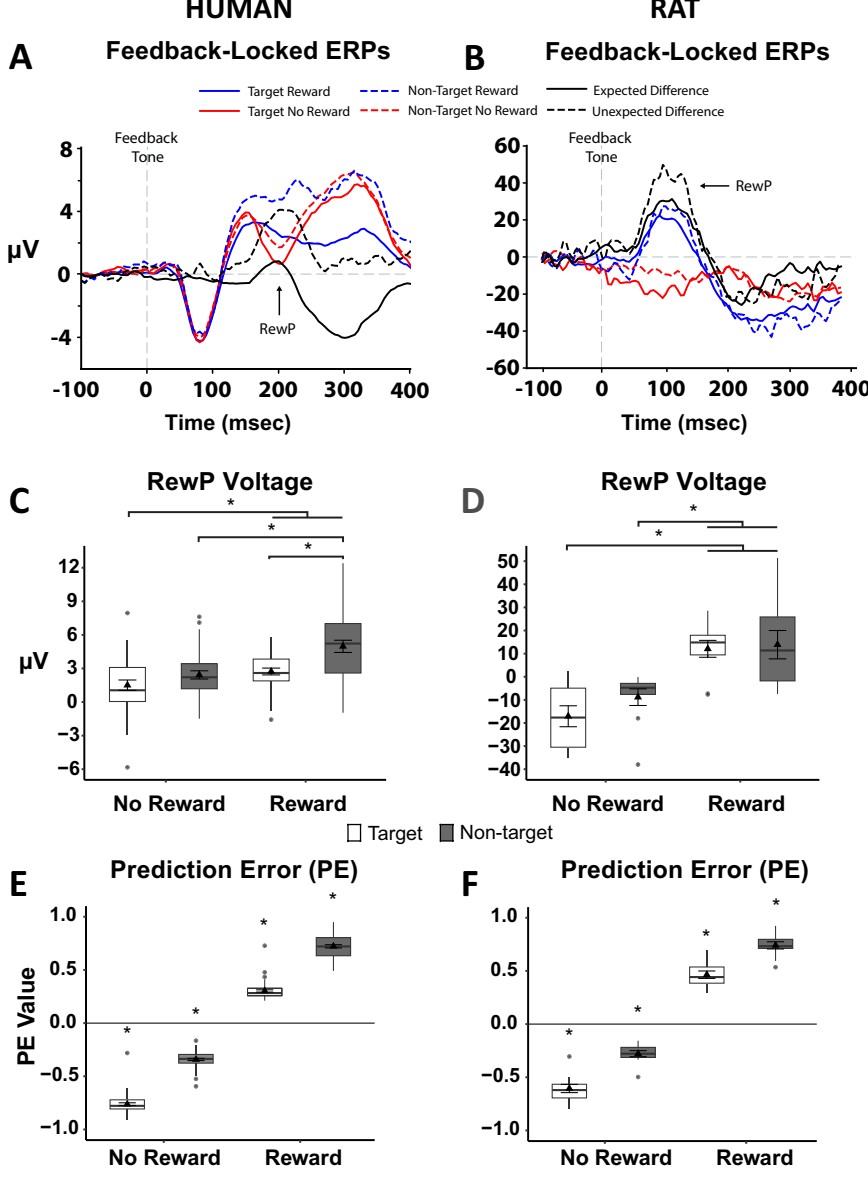

determine the association between the ERP/LFP voltage at each timepoint within each trial and the PE associated with that trial. Notably, in both species (Fig. 3A, B), the regression coefficient peaked in the time window corresponding to the RewP. This indicates that during the RewP period, greater ERP/LFP amplitudes are associated with more positive PEs, whereas lower ERP/LFP amplitudes are associated with less positive PEs. That is, the more positive ERP/LFP for rewarded vs. unrewarded non-target responses appears to reflect the positive PE that occurs when rewards are unexpected. Analogous models using outcome and expected value (i.e., the sub-components of PE) are presented in Supplementary Fig. 5.

To provide additional validation for this regression approach, we used the intercept and regression coefficient generated at each timepoint to predict the ERP/LFP amplitude associated with various PEs. The predicted ERP/LFP traces (Fig. 3C, D) replicated the traces observed in humans and rats (Fig. 2A, B), and, importantly, differences in the predicted traces emerged at the time window corresponding to the RewP. Moreover, the predicted ERP/LFP was greater when the hypothetical PE was more positive. To ensure that the predicted ERP/LFP traces aligned with the activity we recorded from our human or rodent subjects, we plotted the ERP/LFP for trials associated with a PE greater than 0.5 or less than −0.5 (Fig. 3E, F). As expected, and consistent with our predicted ERP/LFP values, more positive

PEs were associated with a more positive ERP/LFP, and this difference was most evident in the RewP time window.

## Modafinil effects on frontal electrophysiological signals and PE values

To understand whether electrophysiological markers of cognitive flexibility are modulated by altering dopamine transmission, we administered the dopamine transport blocker modafinil to a separate cohort of humans (*N* = 29) and the same cohort of rats described above (*N* = 11) prior to PRL testing and EEG recording. Under the placebo/vehicle condition, we successfully replicated the frontal RewP observed between unrewarded and rewarded responses in both humans (Fig. 4A) and rats (Fig. 4B). In humans, a 1-way ANOVA of the ERPs following target responses revealed a significant main effect of Feedback [$F_{(1,26)} = 10.16$, $p = 0.004$] (Fig. 4C). The same significant Feedback effect [$F_{(1,9)} = 15.08$, $p = 0.004$] was identified in the 1-way ANOVA of the rodent LFPs (Fig. 4D). In both species, ERPs/LFPs were more negative following unrewarded target responses relative to rewarded target responses, with, unexpectedly, no effect of modafinil (see Supplementary Fig. 6 for ERPs from all modafinil doses for both species).

We then fitted the same Q-learning algorithm to PRL performance. One-way ANOVA models examining PEs following target responses

**Fig. 3 | The strong relationship between ERP voltage following reward feedback and PE values was comparable between humans (left) and rats (right).** Linear regression revealed a strong positive correlation between ERP voltage during the reward positivity (RewP) period and PE value in both humans (**A**) and rats (**B**). Data reflect average regression coefficient values. We then predicted the ERP response based on several fixed PE values (PE = 0.3, blue dashed line; PE = 0.6, blue solid line; PE = −0.3, red dashed line; PE = −0.6, red solid line) and showed that the predicted ERP was more negative or positive when the PE value was negative or positive, respectively, in both humans (**C**) and rats (**D**). These effects only emerged during the RewP period and were nearly identical to the actual ERP voltages when split between high (PE > 0.5; blue solid line) and low PE (PE < −0.5; red solid line) values in both humans (**E**) and rats (**F**). Data reflect grand average ERP traces.

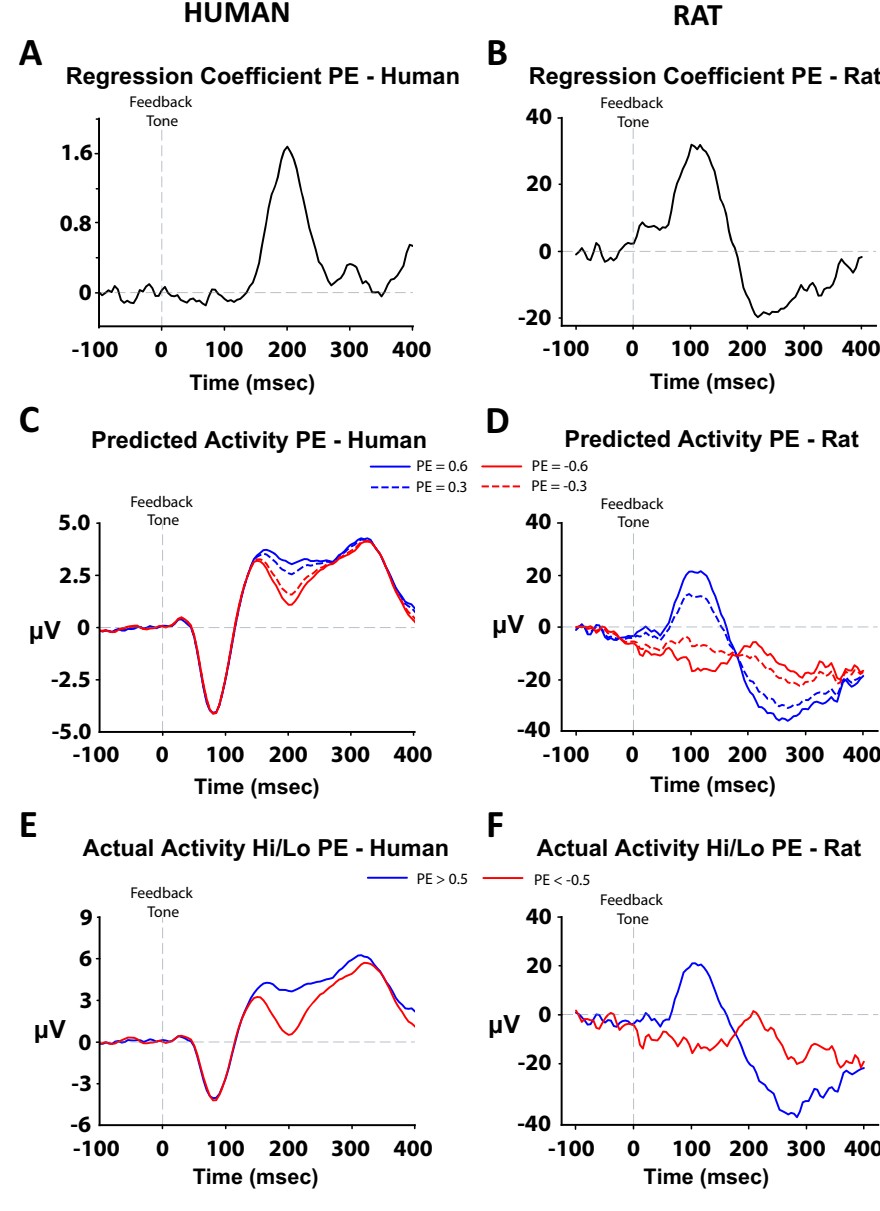

revealed a significant main effect of Feedback for both humans [$F(1,28) = 7630.16$, $p < 0.001$] (Fig. 4E) and rats [$F(1,10) = 2737.01$, $p < 0.001$] (Fig. 4F). Interestingly, in rats, a 1-way ANOVA across all modafinil doses revealed significant linear effects for the rewarded condition [$F(5,50) = 2.62$, $p = 0.035$] with higher doses resulting in more positive PEs; this was not the case for the unrewarded condition [$F(5,50) = 1.90$, $p = 0.111$]. Although the linear effects analysis of human PEs was not significant, the pattern of results was similar to rodents (i.e., increasing PEs with increasing modafinil dose). Although a wide dose response range was used in rodents, matching doses across species is challenging[48] and it is possible that the higher doses used in rats (e.g., 32 and 64 mg/kg) were relatively greater than the highest human dose. Thus, the pattern of increasing PEs across doses may have been more robust if the human participants were exposed to higher modafinil doses.

## Discussion

Using analogous PRL tasks across species, along with equivalent EEG data processing and identical computational analyses of behavioral and EEG data, we found remarkably concordant behavioral, computational, and electrophysiological findings across species, including a neurophysiological response consistent with a RewP in both humans and rats. In humans, the RewP was observed over frontocentral brain regions, while the rodent RewP was recorded directly from the ACC. Notably, in humans, this component peaked approximately 200 ms following feedback, which is earlier than the more typically observed ~300 ms peak[49]. We postulate that this latency shift is likely due to using auditory feedback rather than the more typical visual feedback, but recognize that other task design effects may also be involved in this latency shift. Behavioral performance between species was comparable, as indicated by greater sensitivity to positive feedback (i.e., target win-stay) relative to negative feedback (i.e., target lose-shift), both of which significantly correlated with the overall number of reversals in both species. Although the number of reversals and win-stay responses were generally higher and lose-shift ratios were generally lower in humans compared to rats, these findings are consistent with previously published PRL data in both species[40,50]. Importantly, Q-learning computational analyses revealed that both humans and rats assigned greater value to target vs. non-target stimuli, and these values alternated between response apertures as the criterion for reversals was achieved and reward contingencies were switched. Consistent with our prior finding[51], we found a positive correlation between the beta parameter and the number of completed reversals, likely because a higher beta parameter promotes a greater tendency to exploit the action with the higher value. Taken together, these findings confirm that humans and

**Fig. 4 | Modafinil did not alter feedback-locked ERPs, but did disrupt PE values in rats at high doses.** Testing with the placebo/vehicle produced a similar reward positivity (RewP) as previously shown (Fig. 2) in both humans (**A**) and rats (**B**). Modafinil did not alter the magnitude of the RewP in either species (**C**, **D**). However, positive PE values increased and negative PE values became less negative in response to high doses of modafinil in rats (*n* = 11; **F**). A similar trend may have emerged in humans (*n* = 30; **E**) at higher doses. Grand average ERP traces time-locked to the feedback tone are presented for all trial types (rewarded target = blue solid line; non-rewarded target = red solid line; rewarded non-target = blue dotted line; non-rewarded non-target = red dotted line) and expectancy-based difference waveforms (Expected = Rewarded Target – Non-rewarded Non-Target, and Unexpected = Rewarded Non-Target – Non-rewarded Target). Voltage and PE data are presented using box plots marking the median value (center line), the 25th and 75th percentiles (the outer edges), and ± 1.5 times the interquartile range (the whiskers), as well as the mean (triangle) ± standard error of the mean (error bars).

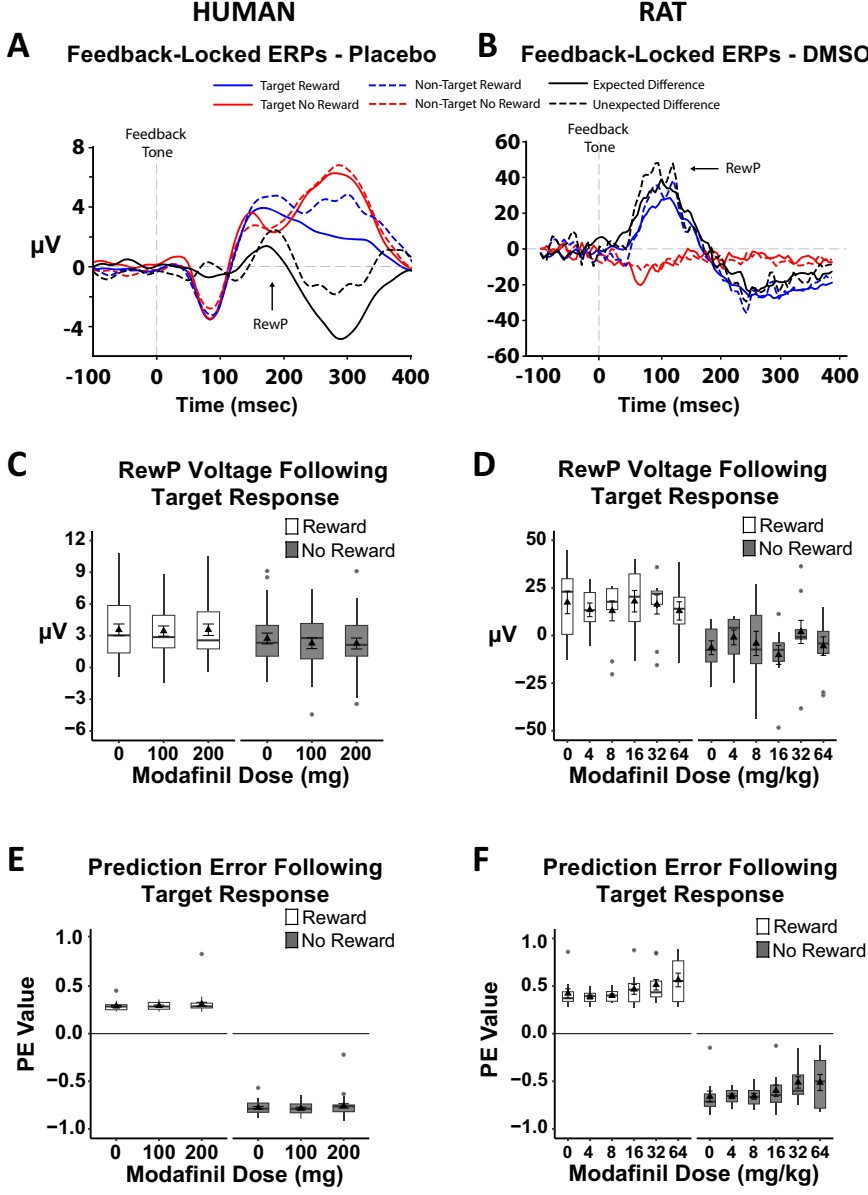

rats performing the PRL task exhibit concordance in both behavioral performance and evoked neurophysiological responses.

Reward PEs are fundamental for learning[52], and signal when an actual outcome deviates from what was expected. As the value assigned to the target was greater than the non-target, one would likely expect a reward after a target response. Importantly, rewarded outcomes elicited positive PEs, whereas non-rewards elicited negative PEs. However, the PE signal following a rewarded non-target response was more positive than rewarded target responses, and a non-rewarded target response elicited a more negative PE than non-rewarded non-target responses. Interestingly, the direction and magnitude of the frontal voltage across the RewP window following feedback were strikingly similar to the direction and magnitude of PEs across all four trial types in both species (Fig. 2C–F), and PEs significantly correlated with voltage during the RewP window in both species as well. These results suggest that the frontal voltage changes triggered by reward feedback represent a neural marker of PEs that is consistent across humans and rats. Less positive deflections after an unexpected reward omission are associated with greater negative PEs, whereas smaller deflections are associated with weaker negative PEs. Blunted negative PEs may contribute to perseverative responding and less flexibility if reward omissions fail to signal that an error has occurred and that alternate options

should be explored. Future research focused on understanding the trial-level relationship between neural activity and choice selection may provide further insight into the role of PEs in cognitive flexibility.

Because midbrain dopaminergic signals to the ACC may contribute to and/or modulate the RewP[26,53], we hypothesized that indirectly enhancing dopamine levels (among other neuromodulators) via modafinil would enhance both the RewP and PEs. Interestingly, in rodents, we found that higher doses of modafinil increased positive PEs for rewarded target responses (i.e., PEs became more positive as modafinil dose increased) but suppressed negative PEs for unrewarded target responses (i.e., PEs became less negative as modafinil dose increased). The effect on PEs may disrupt the ability to properly identify and update the value of the target stimulus, ultimately impacting choice behavior. Notably, we only saw this effect in rodents administered the higher modafinil doses, and animals also completed fewer reversals and exhibited alterations in win-stay/lose-shift strategies (see Supplementary Table 2). The absence of this effect in humans suggests that the maximal human dose administered (200 mg) was not sufficient to disrupt behavior and neurophysiology in a manner consistent with that observed in rats.

It is possible that by increasing dopamine levels, high doses of modafinil might have interfered with the ability of midbrain dopamine neurons

to appropriately suppress activity during an unexpected event, thereby suppressing the PE, which is associated with an attenuated RewP. This is consistent with prior evidence indicating that increased dorsomedial striatum activity reduces the impact of loss, thereby impairing performance in a reversal learning task[54]. Critically, modafinil also modulates other monoamine systems[55], such as noradrenaline and serotonin, which also play a role in regulating reward PEs and value updating[56,57]. Indeed, noradrenergic neurotransmission plays a central role in error monitoring[58]. Thus, it is unclear whether PRL disruptions evident in rats after the highest dose of modafinil are due to alterations in dopamine neurotransmission or another neurotransmitter system. Alternatively, another possible explanation for why modafinil did not modulate PEs and ERPs/LFPs may have to do with the subjects. All human participants were healthy and without any psychiatric conditions, and all rats were naïve and otherwise healthy. Thus, dopamine signaling in healthy subjects may already be optimal, and any additional increase may impede performance. It is possible that modafinil may improve PRL performance in subjects that are otherwise healthy but exhibit reduced task performance or in patient samples characterized by reduced cognitive flexibility. Future studies that are sufficiently powered could group participants based on task performance (i.e., optimal vs poor) before modafinil administration or consider neuropsychiatric samples.

Importantly, the results of the cross-species studies presented here highlight the potential to replicate the rodent study using a parallel patient sample. If successful, this and other similar cross-species approaches may be used to test novel putative pharmacotherapies using similar behavioral and neurophysiological measures with high predictive validity[59]. Such an approach would identify which neural mechanisms linked to behavioral endpoints are conserved across species and thus would be appropriate to assess using an animal model. Importantly, as a major step towards this goal, our findings demonstrate that concordant behavioral, computational, and neurophysiological measures are observed in humans and rodents performing a cross-species task. A similar approach could be used to identify neural mechanisms of and treatments for disorders featuring deficits in cognitive flexibility and other behaviors that can be accurately measured across species and linked to conserved neural processes.

## Methods
A detailed description of the multidisciplinary research program of which the present study was part is described at: clinicaltrials.gov/study/NTC02855229.

### Humans subjects
Sixty-three volunteers, aged 18–45 years, were recruited for the first PRL cohort (which did not include modafinil testing). A total of 54 were retained (19 male, 35 female) for final behavioral data analyses, of which 34 (13 male, 21 female) were retained for EEG analyses after further exclusion due to poor EEG data quality. Thirty separate right-handed volunteers were recruited for the second PRL cohort (which included modafinil), and a total of 29 subjects (14 male, 15 female) were retained for final data analyses. Subjects were free of any psychiatric history, as determined by a clinician-administered Structured Clinical Interview for DSM-5 (SCID-5)[60]. Subjects were compensated $452 for participation. All ethical regulations relevant to human research participants were followed, and all procedures were approved by the Mass General Brigham Institutional Review Board. Subjects provided written informed consent in the presence of a medical doctor prior to participation.

**Human PRL task procedure.** The first cohort participated in a single testing session. Subjects were randomly assigned to either the PRL task or a Flanker task (the results of which were reported separately)[61]. Analyses examining the effects of task order revealed no significant differences. The second PRL cohort completed three sessions, separated by at least one week, using a double-blind, within-subjects, placebo-controlled design; across sessions, subjects were administered 0 mg (placebo), 100 mg, or 200 mg modafinil (2 h pretreatment).

Subjects completed a modified version of the PRL task (Supplementary Fig. 1A) while seated ~70 cm from a computer monitor inside an acoustically and electrically shielded booth. All stimuli were presented on a 22.5-in VIEWPixx monitor (VPixx Technologies, Saint-Bruno, Canada) using PsychoPy software. In this PRL task, participants were tasked to choose between two stimuli, one of which had been randomly designated as the target stimulus at the beginning of the session. Participants received probabilistic feedback in that, if the target stimulus was chosen, a reward would follow 80% of the time. Similarly, if non-target stimuli, negative feedback was given 80% of the time. Thus, spurious feedback would occur 20% of the time. The target or non-target assignment was reversed if the participant selected the target stimulus on 8 consecutive trials irrespective of feedback.

To ensure parallel instructions between species, participants were not made aware of reversing contingencies. Rather, they were instructed that they would need to *"choose between two circles in order to win as much money as you can"* and that they would hear one tone indicating a win/correct selection or a different tone indicating an incorrect selection. Participants completed 10 practice trials to familiarize themselves with the trial structure and the two tones. One session consisted of 300 trials with one break after 150 trials.

Every trial started with the presentation of a fixation cross of random duration between 500 and 1000 ms. Stimuli consisted of a red and blue circle, randomly placed on the left or right side of the screen, presented for maximally 2000 ms or until a response was given. Participants selected the left circle by pressing "c" or the right circle by pressing "m" on a keyboard. As soon as the response was given, a black border appeared around the selected circle for 400 ms. Following a random delay of between 400 and 600 ms, auditory feedback (either a 700 Hz or 1000 Hz pure sine wave) was played for 200 ms. Assignment of the reward and omission outcome to the high or low tone was counterbalanced between participants. If the subject received positive feedback, the feedback tone was followed by the sound of a coin dropping for 1200 ms. This sound was added to mimic the consumption of the food reward in rodents. If participants failed to answer within 2000 ms, a 300 Hz tone was played together with a visual stimulus reading "No response!".

**Human EEG acquisition.** In both PRL cohorts, EEG data were recorded using an actiCHamp amplifier and a 96 Ag/AgCl active electrode actiCAP system (Brain Products GmbH, Gilching, Germany) that used an equi-distant spherical montage and was referenced online to a vertex channel (approximating Cz), with a ground electrode approximately at AFz. Data were digitized at 500 Hz using BrainVision Recorder, and impedances were kept below 35 kΩ.

### Rodent subjects
Eleven adult male ($n = 5$) and female ($n = 6$) Wistar rats were used for both baseline and modafinil experiments (Charles River Laboratories, Wilmington, MA, USA). Animals were pair-housed and food-restricted to 85% of their free-feeding body weight throughout behavioral training. After EEG electrodes were surgically implanted, animals were single-housed for the duration of the experiment and all PRL tests. All rats were housed in a vivarium room with a 12-h reverse light-dark cycle, with lights off between 7:00 AM and 7:00 PM. Rats were monitored daily for signs that would prompt a humane endpoint (e.g., excessive weight loss, inappetence, moribund state, or infection), requiring removal from the study and euthanasia. We have complied with all relevant ethical regulations for animal use; all rodent procedures were approved by the UC San Diego Institutional Animal Care and Use Committee, and were conducted in accordance with guidelines from the National Institute of Health and the Association for Assessment and Accreditation of Laboratory Animal Care.

**Rodent EEG surgery and data acquisition.** Prior to testing, rats were anesthetized with a 2% isoflurane/oxygen vapor mixture and secured on a stereotaxic frame (Kopf Instruments; Tujunga, CA, USA). In order to best approximate human EEG recordings, we implanted three different

electrode types over and within the rats' brains: (1) A 1/8" diameter fine silver disc (Hauser and Miller; St. Louis, MO, USA) soldered to a 0.01" diameter PFA-coated stainless steel wire (#792400; A-M Systems; Sequim, WA, USA) was placed on the surface of the skull immediately rostral to bregma; (2) a stainless steel jeweler's screw soldered to the wire described above was implanted in the skull over the frontal cortex (AP + 3.7 mm, ML ± 2.5 mm) and parietal cortex (AP −4.5 mm, ML ± 4.9 mm); (3) a stainless steel wire described above was inserted into the ACC (AP + 2.7 mm, ML ± 0.8 mm, DV −2.3, from bregma), lateral orbitofrontal cortex, nucleus accumbens shell, caudate nucleus, and primary auditory cortex; notably, electrode implantation angles were unified across animals. Recordings from sites other than the ACC will be reported separately. Reference and ground skull screws were implanted bilaterally over the cerebellum. All electrodes were secured initially with Denmat cement, then completely covered with dental acrylic. The wires from all electrodes were secured with gold pins into an electrode interface board (EIB-16; Neuralynx; Bozeman, MT, USA) that attached to a removable amplifier board during data acquisition. Rats were monitored for at least one-week post-surgery before EEG recording. Unifying electrode coordinates, materials, and implantation angles across animals was done to minimize variability in signal orientation[62,63].

During testing, rats were connected to a 16-channel amplifier board (RHD2132; Intan Technologies; Los Angeles, CA, USA) that transmitted electrophysiological data to a USB interface board connected to a computer running RHD2000 interface software (Intan Technologies). Data were continuously recorded at a 1 kHz sampling rate and filtered between 0.1 and 300 Hz. While LFP data were being continuously collected during the PRL task, TTL event markers were recorded to identify presentation of reward feedback. Audio signals were recorded during testing and connected to the EEG acquisition system to confirm the accuracy of the time-lock between tones and neurophysiological signals.

**Rodent PRL task procedure.** Rats were trained and tested in a Plexiglas operant conditioning box (24 × 30 × 29 cm; Med Associates, St Albans, VT, USA) enclosed in a Faraday cage (Med Associates). It consisted of two retractable levers, a food receptacle positioned between the two levers, a stimulus light above each lever, a speaker above the food receptacle, and a house light placed 2 cm below the ceiling on the opposite wall. Tones were created by an audio generator. All programs and collection of data were done on MED-PC V software (Med Associates, St Albans, VT, USA).

The rodent PRL task was designed to be as similar to a human task as possible (Supplementary Fig. 1B). Briefly, rats responded for one of two colored light stimuli (that were illuminated for up to 5 s) by pressing one of two levers (presented 1 s after illumination) under the two lights. Target responses resulted in positive feedback (100 ms tone, 5 or 15 kHz, counterbalanced) on 80% of trials 500–1000 ms after the response and preceded the delivery of a 45 mg (for male rats) or 20 mg (for female rats) sucrose pellet (5TUT, Test Diet), or negative feedback on 20% of trials (other tone) followed by no pellet. Non-target responses resulted in negative and positive feedback on 80% and 20% of trials, respectively. Eight consecutive target responses, regardless of feedback, resulted in the target stimulus switching to the other light. A "reversal" was recorded when a rat successfully made 8 consecutive target responses after a switch. Rats completed 300 trials per session.

EEG recordings were obtained on the 21st day of testing (i.e., after rats had sufficient time to learn the PRL procedure). After one week, EEG recordings were obtained during testing after administration of one of the following doses of modafinil using a within-subjects Latin-square design: 0, 4, 8, 16, 32, 64 mg/kg. The vehicle for modafinil dosing was DMSO, administered at a volume of 1 ml/kg. There was a minimum one-week washout period between tests.

**Cross-species Task Performance and ERP/LFP derivation**
In both species, in addition to reversals, target win-stay probabilities were calculated as the number of responses repeated after a rewarded target

response divided by the total number of rewarded target responses. Target lose-shift probabilities were calculated as the number of responses not repeated after an unrewarded target response divided by the total number of unrewarded target responses. Q-learning parameters, including Q, PE, alpha, beta, and forget values, were also calculated identically in both species (see Supplementary Methods).

All EEG data were analyzed with BrainVision Analyzer 2.1 (Brain Products GmbH, Gilching, Germany) in the following steps: human data were visually inspected to identify gross muscle activity and artifactual channels, and rodent data were checked for polarity inversions. Following this, data were bandpass filtered from 0.1 (12 dB/oct) to 30 Hz (24 dB/Oct Human cohort 1, 48 dB/oct, Human cohort 2, and Rats) using zero-phase Butterworth IIR filters. Human data were then subjected to independent component analysis to remove eye movement and EKG sources, spherical spline interpolation to replace corrupted channels, and finally re-referenced to the common average. Rodent data were re-referenced to the electrode implanted above the left cerebellum. Processed data for both species were then segmented into −1500 to 2000 ms epochs around the feedback stimulus, and segments were rejected channel-wise as artifact if any of these criteria were met: (1) a voltage exceeding ±75 μV (humans) or ±800 μV (rats); (2) a maximum voltage difference of less than 0.5 μV for more than 100 ms within a trial. Human recordings were also checked against two extra criteria: (1) a voltage step exceeding 50 μV, and (2) a maximum voltage difference of more than 150 μV across 200 ms time intervals within a trial.

In both species, feedback-locked data were segmented into individual epochs spanning from 200 ms before and 600 ms after the feedback tone, baseline-corrected (described below), and averaged. In humans, feedback-locked ERPs were quantified at channel 2 (approximating electrode FCz), and baseline-corrected to the −200 to 0 ms pre-feedback window. The RewP was quantified as the average amplitude between 165 and 225 ms post-feedback. In rats, feedback-locked LFPs were baseline-corrected to the −300 to −100 ms pre-feedback window and quantified at the ACC LFP channel as the average activity across the 60–160 ms post-feedback window.

## Statistics and reproducibility
Across all cohorts (i.e., 1 rodent ($n = 11$) and 2 humans ($n = 34$ and $n = 30$)), the associations between task performance and behavioral parameters were assessed using Pearson's correlation. In turn, between-condition comparisons for all parameters, at both the trial- and session-level, were evaluated using 1- and 2-way ANOVAs. Finally, the association between the ERP/LFP voltage and PE values was evaluated using a series of generalized linear models (GLMs) with a Gaussian distribution and identity link function. The general structure of these models was as follows:

$$E\left(Voltage_{dt}\right) = \beta_0 + \beta_1 PE_t \qquad (1)$$

where $E(Voltage_{dt})$ corresponds to the ERP/LFP amplitude at each data point, $d$, within a given trial, $t$, and $PE_t$ is the signed PE value on that given trial.

## Reporting summary
Further information on research design is available in the Nature Portfolio Reporting Summary linked to this article.

## Data availability
Numerical source data used in the present manuscript are hosted on OSF and available at: osf.io/gxkdv. All other data (i.e., raw EEG recordings) that support the findings of this study are available from the corresponding author upon reasonable request.

## Code availability
The following software were used in preprocessing raw data and subsequent analysis: BrainVision Analyzer (v2.0), Python (v3.7.1), NumPy Python library (v1.21.5), pandas Python Library (v1.1.5), SciPy Python Library (v1.4.1), matplotlib Python library (v3.5.3), statsmodels Python Library

(v0.13.5), IBM SPSS Statistics (v24), and GraphPad Prism (v8). Where possible, Python and R code used in the present analysis are available on OSF at: osf.io/gxkdv. EEG preprocessing templates are available from the corresponding author on reasonable request.

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

## Acknowledgements
D.A.P. discloses support for the research and publication of this work from the National Institute of Mental Health grants UH2 MH109334 and UH3 MH109334. The authors would like to acknowledge the members of our scientific advisory board, Dr. Cindy Ehlers, Dr. Stan Floresco, Dr. Patricio O'Donnell, and Dr. Steven Siegel, for their assistance in the development and execution of these studies. We also thank Ms. Jessica Dally for technical assistance.

## Author contributions
A.D., J.B., W.A.C., V.B.R, S.L., D.A.P. contributed to the study design, A.D., H.S.S., B.D.K., S.L., S.N., M.A.R., M.B., A.M.I., R.L., S.P., E.F.C., G.P.N., J.P.K. conducted the experiments, S.A.B., Z.W., H.P., D.G.D., E.M. performed data analyses, A.D., S.A.B., T.L., H.S.S. wrote the manuscript.

## Competing interests
Over the past 3 years, B.D.K. has had sponsored research agreements with BlackThorn Therapeutics, Compass Pathways, Delix Therapeutics, Engrail Therapeutics, Neurocrine Biosciences, and Takeda Pharmaceuticals. Over the past 3 years, V.B.R. has received consulting fees from Engrail Pharmaceuticals, Jazz Pharmaceuticals, and Cohen Veterans Biosciences. Over the past 3 years, D.A.P. has received consulting fees from Arrowhead Pharmaceuticals, Boehringer Ingelheim, Compass Pathways, Engrail Therapeutics, Neumora Therapeutics, Neurocrine Biosciences, Neuroscience Software, and Takeda; he has received honoraria from the American Psychological Society, Psychonomic Society and Springer (for editorial work) and from Alkermes; he has received research funding from the Bird Foundation, Brain and Behavior Research Foundation, Dana Foundation, DARPA, Millennium Pharmaceuticals, the National Institute of Mental Health, and Wellcome Leap; he has received stock options from Ceretype Neuromedicine, Compass Pathways, Engrail Therapeutics, Neumora Therapeutics, and Neuroscience Software. No funding or any involvement from these entities was used to support the current work, and all views expressed are solely those of the authors. All other authors have no conflicts of interest or relevant disclosures.
