## [Transparent Peer Review file · Communications Biology]

Identification of conserved frontal neurophysiological markers of cognitive flexibility in humans and rats

Corresponding Author: Dr Diego Pizzagalli

Version 0:

Reviewer comments:

Reviewer #1

(Remarks to the Author)

This study develops a rat model of a component of the event related brain potential (ERP) called the feedback-related negativity (FRN) recorded while the animals were engaged in a reversal-learning task, and compares it with the FRN of humans engaged in the same task. Computational modeling of the empirical data indicated that FRN amplitude in both groups was modulated by reward prediction errors, consistent with previous research on this phenomenon. Because dopamine prediction errors have been postulated to drive changes in FRN amplitude, the study also evaluated the rat- and human-FRNs when the subjects were given a dopamine agonist. The authors suggest that the rat model can facilitate future clinical research on reward processing by anterior cingulate cortex (ACC; where the FRN is produced).

This study has a number of important strengths. The experiment is well designed and the results are relatively convincing. Notably, the rat LFPs bears a striking resemblance to the human ERPs. The paper is also well-written. Therefore I believe that the study will make a strong contribution to the literature on translational research. That said, I have a number of questions and concerns.

1. The investigators interpret their FRN results in terms of unexpected negative events, i.e., that the FRN is larger to unexpected errors than to expected errors. For example, the paper states that "Larger negative deflections after an unexpected omission of reward are associated with greater negative PEs, whereas smaller negative deflections are associated with weaker negative PEs" (lines 388-392). However, although this interpretation aligns with early conceptualizations of the FRN from over twenty years ago, it is inconsistent with a growing literature indicating that the component is more sensitive to unexpected rewards than to unexpected errors. More importantly, the interpretation is in fact inconsistent with the investigators' own data. I address these two points in what follows:

1a. Regarding the first point, early studies of the FRN indeed emphasized that the FRN difference wave, created by subtracting positive-feedback ERPs from negative-feedback ERPs, was sensitive to the (un)expectedness of the error (Miltner et al., 1997; Holroyd & Coles, 2002). However, over the past few decades an extended series of studies by my lab and elsewhere has indicated that the FRN difference wave is mainly driven by variance in the neural response to reward feedback rather than to non-reward feedback; for this reason, we recommended a name change from "feedback error-related negativity" to "feedback correct-related positivity" (Holroyd et al., 2008) or simply the "reward positivity" (RewP) (Holroyd et al., 2011; Proudfit, 2014). For example, the negativity to neutral feedback is about as large as the negativity to error feedback (Holroyd et al., 2006); and special populations such as people who abuse addictive substances show a reduced positivity to reward feedback but a normal negativity to error feedback (Baker et al., 2011). These results seem to confirm early speculations (Holroyd, 2004) that the ACC produces a default, negative-going ERP to unexpected task-related events (i.e., the N200 ERP component) that is overridden when that unexpected event has positive valence (Holroyd et al., 2008). All of this to say is that the manuscript does not reflect this more recent, nuanced view of this component, which should at least be acknowledged and discussed.

1b. Regarding the second point, Figure 2C and Figure 2D in fact indicate no significant difference in FRN amplitude between expected and unexpected errors for neither the humans nor the rats. Rather, the differences appear for the unexpected vs. expected rewards for both species. So, the investigators' own data indicate that this ERP component reflects variation in the amplitude of positive PEs rather than negative PEs.

2. For related reasons, the reversal learning task is not well-suited to study the reward positivity because it confounds feedback valence with feedback probability; because errors are relatively infrequent, the larger negativity to the errors could simply be due to the fact that the error feedback is an infrequent event, rather than to it being an infrequent error specifically. Moreover, infrequent events tend to elicit a series of ERP components that overlap both spatially and temporally, which can confound measurement of RewP amplitude. It is therefore essential to rule out this confound when evaluating RewP amplitude. The approach I recommend is to create two difference waves, one for unexpected outcomes and one for expected outcomes:

unexpected RewP = [No Reward, Target] - [Reward, Non-target]

expected RewP = [No-Reward, Non-target] - [Reward, Target]

in which case the unexpected RewP should be larger (more negative) than the Expected RewP. But to demonstrate the validity of this conclusion, the scalp distribution of the difference waves must be checked to ensure that they are frontal-central (and not, for example, posterior, which suggests overlap with the P300) (see, e.g., Holroyd & Krigolson, 2007; Krigolson, 2018).

3. For similar reasons, ERPs for these four conditions (target/non-target as a function of reward/no reward) (c.f. Fig 2c) should be provided. The ERPs following modafinil administration should also be shown.

4. line 589: The particular choice of temporal window for evaluating the human FRN should be justified, especially given that it appears to come after the observed FRN, which peaks at about 200 ms (Fig 3A) (see also the FRN meta-analysis by Sambrook and Goslin, 2015, who recommend specific temporal windows).

5. The computational model represents the two outcomes as either a 1 (good) or a zero (bad). However, the forget parameter tends to force the Q values to zero. If I understand this correctly, then consequently the model learns that any actions that haven't been tried for a while are actually bad rather than neutral (as would be the case if it were simply forgetting). Do the results change if the outcomes range from 1 (good) to -1 (bad) and the q-values are initialized at zero (rather than .5)? Alternatively, do the results change if the values in update-equation #5 are changed to converge .5 when the system forgets?

8. A difficulty with these reward expectancy paradigms is that because the subjects anticipate the outcomes, their behaviors can differ before receiving rewarded vs. non-rewarded feedback. Can the authors provide evidence that the feedback-related ERPs are not confounded by movement artifacts (especially for the rats)?

Minor points:

1. line 173: The authors should clarify that modafinil is a DA agonist.

2. I have never seen intracranial recordings in rats called "ERPs" before. In my opinion it would be less confusing if these are called "LFPs".

3. line 262: The authors should please clarify how "average voltage change" is quantified. Is it the average value within a window?

4. lines 396-411: In this section, it was not immediately clear to me which parts refer to the rat data vs. which refer to the human data.

5. line 423: I appreciate the argument that modafinil may not target cortex, but given that fact, why didn't the authors choose a drug that would do so? For what it is worth, Baker et al. (2016) found genetic evidence that RewP amplitude relates to individual differences in the expression of D4 receptors in ACC.

6. In Figure 1E, the beta values for 3 subjects are all about 10. Does this correspond to the maximum value over which the parameter was optimized?

7. The labels "high PE" vs "low PE" in Figure 2E are ambiguous. I assume these terms correspond to positive RPE vs. negative RPE, but they could also be interpreted as meaning high unsigned PE vs. low unsigned PE. In other words, the component could be larger to larger prediction errors, irrespective of whether the outcome is good or bad.

8 The authors should elaborate on how the model results were used to create the data in Figures 3C and 3D.

9. There is a wider literature on animal models of the RewP/FRN that in my opinion should be acknowledged: e.g., Velozi & Procyk, 2009; Kehrer et al., 2024; Cavanagh et al., 2021, 2022; Hyman et al., 2017

Signed,
Clay Holroyd

References:

Baker, T. E., Stockwell, T., Barnes, G., & Holroyd, C. B. (2011). Individual differences in substance dependence: At the intersection of brain, behavior and cognition. *Addiction Biology*, 16, 458-466.

- Baker, T. E., Stockwell, T., Barnes, G., Haesevoets, R., & Holroyd, C. B. (2016). Reward sensitivity of anterior cingulate cortex as an intermediate phenotype between DRD4-521T and substance misuse. *Journal of Cognitive Neuroscience*, 28, 460-471.
- Cavanagh, J.F., Gregg, D., Light, G. A., Olguin, S., Sharp, R. F., Bismark, A. W., ... Young, J. W. (2021). Electrophysiological biomarkers of behavioral dimensions from cross-species paradigms. *Translational Psychiatry*, 11(482), 1–11.
- Cavanagh, James F., Olguin, S. L., Talledo, J. A., Kotz, J. E., Roberts, B. Z., Nungaray, J. A., ... Brigman, J. L. (2022). Amphetamine alters an EEG marker of reward processing in humans and mice. *Psychopharmacology*, 239(3), 923–933.
- Holroyd, C. B. (2004). A note on the oddball N200 and the feedback ERN. In M. Ullsperger & M. Falkenstein (Eds.), *Errors, conflicts, and the brain: Current opinions on performance monitoring*, (pp. 211-218). Leipzig: MPI of Cognitive Neuroscience.
- Holroyd, C. B., & Coles, M. G. H. (2002). The neural basis of human error processing: Reinforcement learning, dopamine, and the error-related negativity. *Psychological Review*, 109, 679-709.
- Holroyd, C. B., & Krigolson, O. E. (2007). Reward prediction error signals associated with a modified time estimation task. *Psychophysiology*, 44, 913-917.
- Holroyd, C. B., Hajcak, G., & Larsen, J. T. (2006). The good, the bad and the neutral: Electrophysiological responses to feedback stimuli. *Brain Research*, 1105, 93-101.
- Holroyd, C. B., Krigolson, O. E., & Lee, S. (2011). Reward positivity elicited by predictive cues. *Neuroreport*, 22, 249-252.
- Holroyd, C. B., Pakzad-Vaezi, K. L., & Krigolson, O. E. (2008). The feedback correct-related positivity: Sensitivity of the event-related brain potential to unexpected positive feedback. *Psychophysiology*, 45, 688-697.
- Hyman, J. M., Holroyd, C. B., & Seamans, J. K. (2017). A novel neural prediction error found in anterior cingulate cortex ensembles. *Neuron*, 95, 447-456.
- Kehrer, P., Brigman, J. L., & Cavanagh, J. F. (2024). Depth recordings of the mouse homologue of the Reward Positivity. *Cognitive, affective & behavioral neuroscience*, 24(2), 292–301.
- Krigolson O. E. (2018). Event-related brain potentials and the study of reward processing: Methodological considerations. *International journal of psychophysiology : official journal of the International Organization of Psychophysiology*, 132(Pt B), 175–183.
- Miltner, W. H., Braun, C. H., & Coles, M. G. (1997). Event-related brain potentials following incorrect feedback in a time-estimation task: evidence for a "generic" neural system for error detection. *Journal of cognitive neuroscience*, 9(6), 788–798.
- Proudfit G. H. (2015). The reward positivity: from basic research on reward to a biomarker for depression. *Psychophysiology*, 52(4), 449–459.
- Sambrook, T. D., & Goslin, J. (2015). A neural reward prediction error revealed by a meta-analysis of ERPs using great grand averages. *Psychological bulletin*, 141(1), 213–235.
- Vezoli, J., & Procyk, E. (2009). Frontal feedback-related potentials in nonhuman primates: modulation during learning and under haloperidol. *The Journal of neuroscience : the official journal of the Society for Neuroscience*, 29(50), 15675–15683.

Reviewer #2

(Remarks to the Author)

The authors provide evidence that feedback-related electrophysiological signatures are similar between humans and rats, but neither is affected by modafinil. This line of work continues an exciting and important new direction in translational cognitive neuroscience. However I have some concerns about the quantification and interpretation of the 'feedback related negativity' reported here. The rat 'FRN' appears to simply be a reward-related increase that has been subtracted from a flat non-event on loss trials (Fig 2B) This appears to be different from the human FRN - although I admit that there may be evidence for this phenomenon (Fig 3D) yet it is too vague to definitely understand. Together, these issues complicate the interpretation of the main findings, which further complicates the interpretation of modafinil challenge. Fortunately, these are addressable concerns.

While the human FRN is unfortunately non-specifically defined across the literature, the important point is that both the reward and loss electrophysiological responses contain unique information content. Sometimes the human FRN is simply defined as the loss-specific N2 component (which increases with novelty, need for control, and scales with -PE), and other times is referred to as the reward-loss difference which conflates reward-specific and loss/control-specific activities. Figure 3C of the current manuscript shows a compelling test of this pattern: +PE scales positively with this difference wave construct, and -PE scales negatively. This is similar to other previously published studies.

Now, the big question is if the rat ERP difference wave contains the same information content as in humans. Prior reports have detailed +PE electrophysiological responses in rats and in mice, but the electrophysiological response to -PE remains undetermined. In Figure 3D, it appears that there is a similar trend for -PE for rats, but this is difficult to determine for three reasons:

- i) The greyscales are exceedingly hard to parse,
- ii) Fig 3F only shows an effect for 'high PE' and no effect for 'Low PE',
- iii) We don't get a chance to see and rat ERPs from different loss conditions or -PE bins.

Major Points

1) Electrophysiological homologues of -PE: We need to see the rat No Reward ERPs for Target and Non-Target trials as well as the ERPs from different -PE bins. These would help identify what variance is occurring in this time window on loss trials within the context of the broader ERP pattern. In short, does the loss condition contain information content?

2) There is an over-focus on model outcomes and an under-focus on simple performance. The paragraph describing the PE differences between reward and loss trials (pg. 11) shows patterns that should simply be expected if the RL model worked in any way. It should probably be moved to the supplement. Following this rationale, the depiction of drug effects on these PE trends (pg. 13) are not very revealing. It would be more illuminating to simply describe drug effects on manifest aspects of choice selection and accuracy.

3) Drug mechanism of action: Modafinil is described as 'altering dopaminergic transmission'. Most depictions of this drug describe an equally important role of dopamine and norepinephrine.

Minor Points:

1) The term 'cognitive flexibility' is ubiquitous yet under-defined throughout the abstract and introduction. Although it is explained as something akin to task set formation or set shifting, it then becomes interpreted as a feature underlying reversal learning. These are certainly all related to each other, but the progression of the introduction makes it hard for a reader to know what the paper is about until a few pages in.

2) The interpretation of the softmax beta parameter as representing 'exploration' is a little generous. I realize there is a history here, but it was a little too excitable even during its time. The softmax beta is generally a catch-all for any behaviors involved in deterministic selection. Thus larger values should correlate with accuracy (when the task allows), and smaller values would relate to anything involved in non-deterministic selection, which includes exploration but also includes distractibility, forgetting, just pressing buttons, being bad at the task, etc. etc.

Reviewer #3

(Remarks to the Author)

The authors report a cross-species study addressing cognitive flexibility as reflected in a reversal learning task in humans and rats. The study focused on the representation of reward prediction errors (RPEs) in the feedback-related negativity (FRN) of the human frontocentral EEG and an analogous deflection in the local field potentials recorded from rats' anterior cingulate cortex (ACC). A Q-learning model was fit to the behavioral data to estimate trial-wise RPEs. Moreover, the stimulant drug modafinil was administered and effects on behavior and FRN were investigated. Similar results were obtained from both species. Surprisingly, modafinil had no significant effect on the FRN in humans and the FRN analog in rats.

This is an interesting and well-conducted comparative cross-species study demonstrating a feasible and effective example of translational research. Given the relatively large literature on reversal learning in humans and rodents the novelty of the findings is somewhat limited, but in my view the main merit of this study is the close matching of tasks, experimental procedures, and analyses in both species. It nicely demonstrates the methodology of cross-species studies that can help developing valid animal models, e.g., for testing pharmaceuticals and their effects on higher cognitive functions.

I have a few comments and suggestions that can be addressed in minor to moderate revisions:

1. The choice of the tested drug, modafinil, should be justified a bit more extensively. Reducing the effects to dopamine reuptake inhibition seems to simplify the effects of modafinil somewhat too much. In addition to interactions with the DAT, effects on the norepinephrine transporters and alpha1 receptors seem to exist. The net effect of modafinil administration appears to be an increase of dopaminergic, noradrenergic and even serotonergic activity. In the discussion of the (almost non-existent) effects of modafinil on the FRN and model-derived RPEs the authors should refer to previous pharmacological studies, even though they mostly addressed the ERN rather than the FRN, e.g., de Bruijn et al., *Psychopharmacology*, 2004; Riba et al., *J Neurosci*, 2005.

2. In the introduction (line 152) and discussion (line 393) the role of DA in the generation of the ERN/FRN based on theory

by Holroyd and Coles (2002) is discussed. Several authors have challenged the view that DA signals (in particular dips in DA release upon negative RPEs) could drive the generation of these ERPs. A quite comprehensive discussion of this issue can be found, for example, in Ullsperger et al. Trends Cogn Sci 2014. I suggest to tone down the according sentences. In my view, DA for sure modulates the performance monitoring and the related ERPs, but this may happen indirectly via striatal and/or tonic cortical effects. The slowness of the DA signal and DA clearance from the synaptic clefts in cortex renders it very unlikely that a dopaminergic RPE signal is the main driving force of the ERN or FRN.

3. From the analyses it is not directly clear that the FRN is an axiomatic RPE signal; it could be driven just by the outcome. It appears important to demonstrate that the expected value is also reflected in the FRN (cf. Caplin & Dean, Curr Opin Neurobiol, 2008; Rutledge et al., J Neurosci, 2010). A simple way to address is to use both constituents of the RPE, outcome and expected value, as predictors in the regression analysis (see, for an example, Fischer & Ullsperger, Neuron, 2013). Results of such an analysis should be reported at least in the supplements.

4. Please elaborate a bit more on the LFP recordings. What was the reference? How was polarity of the LFPs matched across individual animals?

5. Some references seem not to fit to the claims that were made. E.g., in line 146, the paper by Falkenstein et al., 1991, is on the ERN rather than the FRN and not related to unexpected negative vs. positive outcomes. Similarly, Carter et al. (1998) does not address the sources of the FRN. A paper that might fit much better is Hauser et al., Neuroimage, 2014, where a simultaneous EEG/fMRI study on the FRN is reported.

6. It would have been nice to see some (supplementary) figures on model recovery and posterior predictive checks of the selected model.

7. The FRN is interpreted as the difference wave of negative minus positive outcomes. It might be worth mentioning that, in rats, there is actually no visible deflection for negative outcomes and that the FRN-like difference wave is actually driven by a deflection for positive outcomes.

8. It would strengthen the paper massively, if a closer link between ERPs and behavioral adjustments could be shown. In other probabilistic learning tasks (including reversal learning), the P300/centroparietal positivity elicited by the feedback predicted future choice behavior (e.g., Fischer & Ullsperger, Neuron, 2013; Kirschner et al., Neuroimage, 2022). Can anything similar be found in the human, and even more interestingly, rat data?

Version 1:

Reviewer comments:

Reviewer #1

(Remarks to the Author)

The authors' revisions have thoroughly addressed nearly all of my concerns about the previous version of the manuscript. I have only a few remaining comments:

1. The revised manuscript now reports that the reward positivity is evaluated in a 145-200 ms window post-feedback. I wonder why the authors chose this window? It is not the typical time period to observe the reward positivity, and in fact their own data suggest that the RewP peaks at 200 ms (see, e.g., Fig 3A). In fact, the reward positivity is normally seen around 240-340 ms, so why does it occur so early in this study? Is it because of the use of auditory feedback rather than visual feedback (resulting in faster latency)?

2. Fig 2A and 2B plot the target difference wave and the non-target difference wave, but these are not standard ways of assessing the reward positivity because they confound valence with expectancy. For example, the target difference wave compares a frequent reward with an infrequent no-reward, so the difference can be due to frequency rather than to reward. By contrast, Figure A in the authors' cover letter depicts the expected and unexpected difference waves, which are the more standard way of measuring the RewP because these comparisons remove the probability confound. I recommend that Fig 2A and 2B plot the expected and unexpected difference waves rather than the target and non-target difference waves. Parenthetically, the caption to Fig 2A and Fig 2B should describe the legend.

Reviewer #2

(Remarks to the Author)

The authors have successfully addressed all of my prior concerns.

One minor note: In Figure 2B, there is no "Target Difference" solid black line. Furthermore, the RewP label and arrow appears to be pointing to the red negative deflection, which is confusing.

Reviewer #3

(Remarks to the Author)

The authors provide a substantially revised version of their manuscript in which they addressed all reviewers' concerns. I think the manuscript is now ready for publication. I have no further suggestions or comments.

Version 2:

Reviewer comments:

Reviewer #1

(Remarks to the Author)

The authors have thoroughly addressed my concerns. My only remaining suggestion is that the authors should comment on the early latency of the RewP in the discussion section. This is a rather peculiar finding that I suspect has more to do with the task design than with the use of auditory feedback (though it could be a combination of both). Because latency is a key signature for identifying a component, I believe it important that this discrepancy be acknowledged.

Reviewer 1:

- 1. The investigators interpret their FRN results in terms of unexpected negative events, i.e., that the FRN is larger to unexpected errors than to expected errors. For example, the paper states that “*Larger negative deflections after an unexpected omission of reward are associated with greater negative PEs, whereas smaller negative deflections are associated with weaker negative PEs*” (lines 388-392). However, although this interpretation aligns with early conceptualizations of the FRN from over twenty years ago, it is inconsistent with a growing literature indicating that the component is more sensitive to unexpected rewards than to unexpected errors. More importantly, the interpretation is in fact inconsistent with the investigators’ own data. I address these two points in what follows:**
 - a. Regarding the first point, early studies of the FRN indeed emphasized that the FRN difference wave, created by subtracting positive-feedback ERPs from negative-feedback ERPs, was sensitive to the (un)expectedness of the error (Miltner et al., 1997; Holroyd & Coles, 2002). However, over the past few decades an extended series of studies by my lab and elsewhere has indicated that the FRN difference wave is mainly driven by variance in the neural response to reward feedback rather than to non-reward feedback; for this reason, we recommended a name change from "feedback error-related negativity" to "feedback correct-related positivity" (Holroyd et al., 2008) or simply the "reward positivity" (RewP) (Holroyd et al., 2011; Proudfit, 2014). For example, the negativity to neutral feedback is about as large as the negativity to error feedback (Holroyd et al., 2006); and special populations such as people who abuse addictive substances show a reduced positivity to reward feedback but a normal negativity to error feedback (Baker et al., 2011). These results seem to confirm early speculations (Holroyd, 2004) that the ACC produces a default, negative-going ERP to unexpected task-related events (i.e., the N200 ERP component) that is overridden when that unexpected event has positive valence (Holroyd et al., 2008). All of this to say is that the manuscript does not reflect this more recent, nuanced view of this component, which should at least be acknowledged and discussed.**
 - b. Regarding the second point, Figure 2C and Figure 2D in fact indicate no significant difference in FRN amplitude between expected and unexpected errors for neither the humans nor the rats. Rather, the differences appear for the unexpected vs. expected rewards for both species. So, the investigators’ own data indicate that this ERP component reflects variation in the amplitude of positive PEs rather than negative PEs.**

We appreciate the reviewer's constructive comments and their deep expertise regarding the FRN/RewP; to address this important point, we have amended our conceptualization of the evoked signal throughout the manuscript to align with the RewP framework. Additionally, we also want to note that any relevant difference waveforms included in the manuscript (e.g., Figures 2 and 4) have now been replotted to be the Reward – Non-Reward difference (rather than the Non-Reward – Reward difference, as it was previously), which better aligns with this new framing.

2. For related reasons, the reversal learning task is not well-suited to study the reward positivity because it confounds feedback valence with feedback probability; because errors are relatively infrequent, the larger negativity to the errors could simply be due to the fact that the error feedback is an infrequent event, rather than to it being an infrequent error specifically. Moreover, infrequent events tend to elicit a series of ERP components that overlap both spatially and temporally, which can confound measurement of RewP amplitude. It is therefore essential to rule out this confound when evaluating RewP amplitude. The approach I recommend is to create two difference waves, one for unexpected outcomes and one for expected outcomes:

unexpected RewP = [No Reward, Target] - [Reward, Non-target]

expected RewP = [No-Reward,Non-target] - [Reward,Target]

in which case the unexpected RewP should be larger (more negative) than the Expected RewP. But to demonstrate the validity of this conclusion, the scalp distribution of the difference waves must be checked to ensure that they are frontal-central (and not, for example, posterior, which suggests overlap with the P300) (see, e.g., Holroyd & Krigolson, 2007; Krigolson, 2018).

The reviewer makes an excellent point regarding the unbalanced frequency of outcomes and associated feedback in the reversal learning task. Following their suggestion, we generated the expectancy-based waveforms for all cohorts of humans and rats. These difference waveforms are presented below in Figures A (Study 1) and B (Study 2 – Modafinil); both figures also include the associated scalp topographies for the human wave forms.

In line with the reviewer's expectation, we see that, generally, the Unexpected waveform has a greater magnitude than the Expected waveform for both human ERP and rodent LFP recordings. Moreover, as the reviewer noted, in humans this signal was generally localized to frontocentral regions.

Figure A – Expectancy-based ERP waveforms in humans and LFP waveforms in rodents from Study 1.

Note: Plots A presents the human ERP waveforms and associated scalp topographies, and Plot B presents the rodent LFP waveforms. Across all plots, expectancy waveforms were calculated using four different ERP/LFP waves derived from the reversal learning task and were calculated as: Expected = Non-Rewarded Non-Target – Rewarded Target; Unexpected = Non-Reward Target – Rewarded Non-Target.

Figure B – Expectancy based ERP waveforms in humans and LFP waveforms in rodents across doses of modafinil.

Note: Plots A, B, and C present the human ERP waveforms and associated scalp topographies; Plots D-I present the rodent FLP waveforms. Across all plots, expectancy waveforms were calculated using four different ERP/LFP waves derived from the reversal learning task and were calculated as: Expected = Non-Rewarded Non-Target – Rewarded Target; Unexpected = Non-Reward Target – Rewarded Non-Target.

- 3. For similar reasons, ERPs for these four conditions (target/non-target as a function of reward/no reward) (c.f. Fig 2c) should be provided. The ERPs following modafinil administration should also be shown.**

As recommended by the reviewer, we have amended Figures 2 and 4 to present the evoked signals for both species across all four trial types. Furthermore, we have also separately plotted the evoked signals for each of the dose conditions from the Modafinil study and these can be found in Supplemental Figure 1.

- 4. line 589: The particular choice of temporal window for evaluating the human FRN should be justified, especially given that it appears to come after the observed FRN, which peaks at about 200 ms (Fig 3A) (see also the FRN meta-analysis by Sambrook and Goslin, 2015, who recommend specific temporal windows).**

We appreciate the reviewer's attention to detail and have corrected the description of the temporal window used for the RewP.

- 5. The computational model represents the two outcomes as either a 1 (good) or a zero (bad). However, the forget parameter tends to force the Q values to zero. If I understand this correctly, then consequently the model learns that any actions that haven't been tried for a while are actually bad rather than neutral (as would be the case if it were simply forgetting). Do the results change if the outcomes range from 1 (good) to -1 (bad) and the q-values are initialized at zero (rather than .5)? Alternatively, do the results change if the values in update-equation #5 are changed to converge .5 when the system forgets?**

Thank you for these suggestions. We fitted a modified model where the Q values were initialized at zero. There was no difference between model fits when the Q values were initialized to zero or 0.5 (BIC = 341.92 or 341.83, respectively). In addition, when we modified the model to force the value for the unchosen action to converge to 0.5 resulted in a worse model fit (BIC = 358.55). Importantly, since the PRL task has only two options, if the left action is good then the right action is bad (and not neutral). We and others have explored this possibility previously using "double-update" models, whereby the PE is used to update the value of the chosen and unchosen actions (Barnes et al., 2023; Schlagenhaut et al., 2014); equation 4 in supplement of the current manuscript). The model including the forget parameter was a better fit for our data than the double-update model, and likely improved the model fit by exaggerating the value difference between actions.

6. A difficulty with these reward expectancy paradigms is that because the subjects anticipate the outcomes, their behaviors can differ before receiving rewarded vs. non-rewarded feedback. Can the authors provide evidence that the feedback-related ERPs are not confounded by movement artifacts (especially for the rats)?

We believe it is unlikely that our feedback-related ERPs are confounded by movement artifacts. In humans, the participants were given the standard instruction to remain as still as possible and minimize head movement while completing the task. For rodents, LFPs were referenced to a skull-screw that was implanted directly above the cerebellum, and the activity traces were baseline-corrected using a time-window before the tone was presented; these steps would capture and account for at least some motor related activity. Moreover, anecdotally, we observed rats moving towards the reward port after any response regardless of the trial outcome. Thus, the level of motion evident during rewarded and non-rewarded trials was comparable, yet we observed reward-evoked changes in LFPs. As such, we suggest that it is unlikely that these signals were due to motion artifacts.

7. line 173: The authors should clarify that modafinil is a DA agonist.

As suggested, we have clarified this line to indicate that modafinil is an indirect dopamine agonist.

8. I have never seen intracranial recordings in rats called “ERPs” before. In my opinion it would be less confusing if these are called “LFPs”.

We had hoped that referring to the evoked signals as ERPs for both species would provide uniformity and accessibility to the manuscript. However, we recognize that it also has the potential to be confusing and as suggested have edited the manuscript to refer to the rodent intracranial recordings as local field potentials or LFPs.

9. line 262: The authors should please clarify how “average voltage change” is quantified. Is it the average value within a window?

We have clarified line 262 to refer to the difference in the average voltage across the measurement window, rather than the change in voltage, between rewarded and unrewarded target and non-target trials.

10. lines 396-411: In this section, it was not immediately clear to me which parts refer to the rat data vs. which refer to the human data.

We appreciate the reviewer’s comment regarding the clarity of this section of our discussion. As suggested, we have refined the relevant text to indicate which species

we are referring to when discussing our results.

11. line 423: I appreciate the argument that modafinil may not target cortex, but given that fact, why didn't the authors choose a drug that would do so? For what it is worth, Baker et al. (2016) found genetic evidence that RewP amplitude relates to individual differences in the expression of D4 receptors in ACC.

We have revised the discussion paragraph that describes potential explanations for the minimal effect of modafinil on PRL performance in humans.

12. In Figure 1E, the beta values for 3 subjects are all about 10. Does this correspond to the maximum value over which the parameter was optimized?

The beta parameter was bounded between 0 and 10, which was suggested by Zhang et al. (2020) to be a reasonable range to avoid unstable model estimation.

13. The labels "high PE" vs "low PE" in Figure 2E are ambiguous. I assume these terms correspond to positive RPE vs. negative RPE, but they could also be interpreted as meaning high unsigned PE vs. low unsigned PE. In other words, the component could be larger to larger prediction errors, irrespective of whether the outcome is good or bad.

This assumption was correct. High and low PEs in Figure 3E correspond to positive vs. negative PEs (>0.5 or <-0.5 , respectively). We have modified the text to avoid any ambiguity.

14. The authors should elaborate on how the model results were used to create the data in Figures 3C and 3D.

The model (Figure 3A and 3B) generated an intercept and beta coefficient for each point across the recording window. The coefficient was multiplied by a hypothetical PE value (e.g., 0.75) and the corresponding intercept was added. This process was repeated for each coefficient/intercept value obtained across the recording window, and for a range of potential PE values (e.g., 0.75 to -0.75). We have provided an expanded description in the revised manuscript.

15. There is a wider literature on animal models of the RewP/FRN that in my opinion should be acknowledged: e.g., Velozi & Procyk, 2009; Kehrer et al., 2024; Cavanagh et al., 2021, 2022; Hyman et al., 2017

We agree with the reviewer and have amended our manuscript, where relevant, to refer to this wider animal literature. Thank you for the excellent suggestions.

Reviewer 2:

- 1. The authors provide evidence that feedback-related electrophysiological signatures are similar between humans and rats, but neither is affected by modafinil. This line of work continues an exciting and important new direction in translational cognitive neuroscience. However I have some concerns about the quantification and interpretation of the ‘feedback related negativity’ reported here. The rat ‘FRN’ appears to simply be a reward-related increase that has been subtracted from a flat non-event on loss trials (Fig 2B) This appears to be different from the human FRN - although I admit that there may be evidence for this phenomenon (Fig 3D) yet it is too vague to definitely understand. Together, these issues complicate the interpretation of the main findings, which further complicates the interpretation of modafinil challenge. Fortunately, these are addressable concerns.**

While the human FRN is unfortunately non-specifically defined across the literature, the important point is that both the reward and loss electrophysiological responses contain unique information content. Sometimes the human FRN is simply defined as the loss-specific N2 component (which increases with novelty, need for control, and scales with -PE), and other times is referred to as the reward-loss difference which conflates reward-specific and loss/control-specific activities. Figure 3C of the current manuscript shows a compelling test of this pattern: +PE scales positively with this difference wave construct, and -PE scales negatively. This is similar to other previously published studies.

Now, the big question is if the rat ERP difference wave contains the same information content as in humans. Prior reports have detailed +PE electrophysiological responses in rats and in mice, but the electrophysiological response to -PE remains undetermined. In Figure 3D, it appears that there is a similar trend for -PE for rats, but this is difficult to determine for three reasons:

- i. The greyscales are exceedingly hard to parse,**
- ii. Fig 3F only shows an effect for ‘high PE’ and no effect for ‘Low PE’,**
- iii. We don’t get a chance to see and rat ERPs from different loss conditions or -PE bins.**

We thank the reviewer for their insightful comments. We have modified the formatting of Figure 3 to improve the visual clarity. Figure 3E and F show activity for positive and negative PEs (and we modified the descriptions to remove any ambiguity).

Beyond this, edit, we also wish to highlight that in response to Reviewer 1 we have adjusted the conceptualization of what we previously labelled the FRN to align with the Reward-positivity (RewP) framework that has more recently emerged (see comment 1 from Reviewer 1). This is important as the updated conceptualization of the RewP would suggest that the nature of the signal is driven by a reward-related increase in the signal, and that the non-reward response/signal is the “default” action. With this in mind, we would suggest that the observed rodent LFP signals are entirely in line with what is expected, i.e., a reward-related increase. That said, the entirely absent non-reward response in rodents does present a species-specific behavioral difference that may be meaningful for future research.

Finally, to the reviewer’s third point, the design of the PRL task in the present study does not have an explicit loss condition, rather, it has a “neutral” non-reward condition (noting that there is a place to interpret this as a relative loss outcome). Accordingly, while the reviewer raises the very valid point of reward and loss representing separate unique processes, we are unable demonstrate an electrophysiological response to loss. It is similarly worth noting that the negative PE does not necessarily represent a loss, just that the reward received (i.e., no reward) was less than the expected reward (i.e., some reward). However, Figure 2 now reports ERPs and LFPs for each of the four possible outcomes in the PRL task.

2. Electrophysiological homologues of -PE: We need to see the rat No Reward ERPs for Target and Non-Target trials as well as the ERPs from different -PE bins. These would help identify what variance is occurring in this time window on loss trials within the context of the broader ERP pattern. In short, does the loss condition contain information content?

In response to the reviewer’s comment, we have edited Figures 2A and 2B to present the ERPs for all four trial types (i.e., Rewarded or Non-Rewarded Target or Non-Target). Moreover, we also edited Figures 3E and 3F to provide a more detailed view of observed activity across both positive and negative PE bins.

As per our response to the previous comment, we again wish to highlight that the design of the PRL task in the present study does not have an explicit loss condition rather it has a non-reward condition.

3. There is an over-focus on model outcomes and an under-focus on simple performance. The paragraph describing the PE differences between reward and loss trials (pg. 11) shows patterns that should simply be expected if the RL model worked in any way. It should probably be moved to the supplement. Following this rationale, the depiction of drug effects on these PE trends (pg.

13) are not very revealing. It would be more illuminating to simply describe drug effects on manifest aspects of choice selection and accuracy.

We thank the reviewer for highlighting this. The major focus of our paper was the cross-species validation of the task and neurophysiological readout. Nonetheless, we now include in the supplement how modafinil affected the basic behavioral parameters in both humans and rodents.

4. Drug mechanism of action: Modafinil is described as ‘altering dopaminergic transmission’. Most depictions of this drug describe an equally important role of dopamine and norepinephrine.

In line with the reviewer’s comment and a similar related comment from Reviewer 1, we have revised our description of modafinil to indicate that it’s mechanism of action is not exclusive to dopamine.

5. The term ‘cognitive flexibility’ is ubiquitous yet under-defined throughout the abstract and introduction. Although it is explained as something akin to task set formation or set shifting, it then becomes interpreted as a feature underlying reversal learning. These are certainly all related to each other, but the progression of the introduction makes it hard for a reader to know what the paper is about until a few pages in.

We appreciate the reviewer’s comment and in response have amended both the introduction and abstract of the manuscript to operationalize the term ‘cognitive flexibility’ more clearly and earlier in the manuscript.

6. The interpretation of the softmax beta parameter as representing ‘exploration’ is a little generous. I realize there is a history here, but it was a little too excitable even during its time. The softmax beta is generally a catch-all for any behaviors involved in deterministic selection. Thus larger values should correlate with accuracy (when the task allows), and smaller values would relate to anything involved in non-deterministic selection, which includes exploration but also includes distractibility, forgetting, just pressing buttons, being bad at the task, etc. etc.

As recommended, we have tempered the interpretation regarding a reduction in the beta parameter corresponding to an increase in exploration.

Reviewer #3:

1. The choice of the tested drug, modafinil, should be justified a bit more extensively. Reducing the effects to dopamine reuptake inhibition seems to

simplify the effects of modafinil somewhat too much. In addition to interactions with the DAT, effects on the norepinephrine transporters and alpha1 receptors seem to exist. The net effect of modafinil administration appears to be an increase of dopaminergic, noradrenergic and even serotonergic activity. In the discussion of the (almost non-existent) effects of modafinil on the FRN and model-derived RPEs the authors should refer to previous pharmacological studies, even though they mostly addressed the ERN rather than the FRN, e.g., de Bruijn et al., *Psychopharmacology*, 2004; Riba et al., *J Neurosci*, 2005.

The reviewer is correct in that condensing the effects of modafinil to purely dopamine reuptake inhibition is an oversimplification. In response to this comment, and those of both Reviewer 1 and 2, we have revised our description of modafinil to make note of this. Our revised discussion now highlights that high doses of modafinil may exert effects via dopamine, noradrenaline, or serotonin.

- 2. In the introduction (line 152) and discussion (line 393) the role of DA in the generation of the ERN/FRN based on theory by Holroyd and Coles (2002) is discussed. Several authors have challenged the view that DA signals (in particular dips in DA release upon negative RPEs) could drive the generation of these ERPs. A quite comprehensive discussion of this issue can be found, for example, in Ullsperger et al. *Trends Cogn Sci* 2014. I suggest to tone down the according sentences. In my view, DA for sure modulates the performance monitoring and the related ERPs, but this may happen indirectly via striatal and/or tonic cortical effects. The slowness of the DA signal and DA clearance from the synaptic clefts in cortex renders it very unlikely that a dopaminergic RPE signal is the main driving force of the ERN or FRN.**

We appreciate the reviewer's note regarding the generation of the FRN (now referred to as the RewP) signal and, as suggested by the Reviewer, we have toned down the relevant sentences regarding the role of DA in the generation of these signals.

- 3. From the analyses it is not directly clear that the FRN is an axiomatic RPE signal; it could be driven just by the outcome. It appears important to demonstrate that the expected value is also reflected in the FRN (cf. Caplin & Dean, *Curr Opin Neurobiol*, 2008; Rutledge et al., *J Neurosci*, 2010). A simple way to address is to use both constituents of the RPE, outcome and expected value, as predictors in the regression analysis (see, for an example, Fischer & Ullsperger, *Neuron*, 2013). Results of such an analysis should be reported at least in the supplements.**

We thank the reviewer for this comment. Our revised manuscript now includes an additional regression model that uses the outcome received and the expected value as model predictors. For both species, there is a positive relationship with outcome and a negative relationship with value that corresponds to the ERP window that we had selected. These effects demonstrate that these effects are not simply driven by the valence of the outcome but likely reflect a reward prediction error.

4. Please elaborate a bit more on the LFP recordings. What was the reference? How was polarity of the LFPs matched across individual animals?

As requested, we have included additional details regarding the rodent LFP recordings in the methods section of the manuscript. To briefly reply to the two questions asked by the reviewer:

- 1) The reference electrode was a skull screw implanted over the cerebellum.
- 2) To ensure signal orientation and the polarity of the LFP recordings, we used an implantation/surgical procedure that aligns with recommendations from LFP-literature that emphasizes the impact of electrode geometry and reference choice on signal (Buzsáki et al., 2012; Einevoll et al., 2013), and uses the same stereotaxic coordinates for the ACC electrode, reference electrode location, implantation angles, electrode materials, and recording hardware for each animal. Moreover, we visually evaluated the LFP recordings and averaged waveforms and observed no polarity inversions.

5. Some references seem not to fit to the claims that were made. E.g., in line 146, the paper by Falkenstein et al., 1991, is on the ERN rather than the FRN and not related to unexpected negative vs. positive outcomes. Similarly, Carter et al. (1998) does not address the sources of the FRN. A paper that might fit much better is Hauser et al., Neuroimage, 2014, where a simultaneous EEG/fMRI study on the FRN is reported.

We appreciate the reviewer's comment and the suggested replacement citation. We have amended the relevant citations and further checked the remainder of our citations to ensure their relevance.

6. It would have been nice to see some (supplementary) figures on model recovery and posterior predictive checks of the selected model.

Posterior predictive checks were presented in Supplemental Figure 2. We now also include parameter recovery data in Supplemental Figure 2 demonstrating that the model parameters were recoverable.

- 7. The FRN is interpreted as the difference wave of negative minus positive outcomes. It might be worth mentioning that, in rats, there is actually no visible deflection for negative outcomes and that the FRN-like difference wave is actually driven by a deflection for positive outcomes.**

The reviewer raises a very relevant point; it appears the absence of response to a negative (i.e., non-rewarded outcome) describes a species-specific difference in the reward responsiveness, and we now make note of this in our discussion. Furthermore, we wanted to highlight that, in response to comments from Reviewer 1 we have amended our instantiation of the FRN to be in line with the more recent RewP conceptualization of this waveform (Holroyd et al., 2008; Proudfit, 2015), which argues and provides evidence indicating that this component is indeed driven by a positive deflection following rewarded outcomes.

- 8. It would strengthen the paper massively, if a closer link between ERPs and behavioral adjustments could be shown. In other probabilistic learning tasks (including reversal learning), the P300/centroparietal positivity elicited by the feedback predicted future choice behavior (e.g., Fischer & Ullsperger, Neuron, 2013; Kirschner et al., Neuroimage, 2022). Can anything similar be found in the human, and even more interestingly, rat data?**

The reviewer raises an important point about the nature of learning tasks and how they can be best examined given their iterative nature. Trial-level analysis focusing on current or previous trial parameters predicting the behavior of the current or following trial is increasingly being reported, especially as multi-level and mixed effects models become more prominent in psychophysiological research. However, for the present manuscript, we feel that such an analysis is beyond its scope which aimed to evaluate the neurophysiological signatures of reward across species. We do hope to conduct such an analysis in the future and have included a comment on this approach in the discussion.

That said, we believe that the results in the section entitled '*Relationships between behavioral and electrophysiological measures of PEs*' do point at the link between the neural and behavioral aspects of the PRL task. In this analysis, we used generalized linear models (GLMs) with a gaussian distribution and identity link function to evaluate the association between the ERP/LFP voltage at each timepoint within a trial and the corresponding trial-level PE values. The general structure of these models was as follows:

$$E(\text{Voltage}_{dt}) = \beta_0 + \beta_1 PE_t$$

where $E(\text{Voltage}_{dt})$ corresponds to the ERP/LFP amplitude at each data point, d , within a given trial, t , and PE_t is the signed PE value on that given trial. This structure is

similar to the GLMs of Fischer & Ullsperger (2013), and Kirschner et al. (2022); although notably less complex due to differences in study design. We also want to note that following the first review of our manuscript, we have since supplemented this analysis with models that used trial-level outcome and expected value (i.e., the subcomponents of a PE) rather than the trial-level PE. This model structure would be:

$$E(\text{Voltage}_{dt}) = \beta_0 + \beta_1 \text{Outcome}_t + \text{ExpectedValue}_t$$

Together these models provide a more nuanced picture of the relationship between behavior and the ERPs and a summary of these models and relevant plots can be found in Supplemental Figure 5.

References

- Barnes, S. A., Dillon, D. G., Young, J. W., Thomas, M. L., Faget, L., Yoo, J. H., Der-Avakian, A., Hnasko, T. S., Geyer, M. A., & Ramanathan, D. S. (2023). Modulation of ventromedial orbitofrontal cortical glutamatergic activity affects the explore-exploit balance and influences value-based decision-making. *Cerebral Cortex*, *33*(10), 5783–5796. <https://doi.org/10.1093/cercor/bhac459>
- Buzsáki, G., Anastassiou, C. A., & Koch, C. (2012). The origin of extracellular fields and currents — EEG, ECoG, LFP and spikes. *Nature Reviews Neuroscience*, *13*(6), 407–420. <https://doi.org/10.1038/nrn3241>
- Einevoll, G. T., Kayser, C., Logothetis, N. K., & Panzeri, S. (2013). Modelling and analysis of local field potentials for studying the function of cortical circuits. *Nature Reviews Neuroscience*, *14*(11), 770–785. <https://doi.org/10.1038/nrn3599>
- Fischer, A. G., & Ullsperger, M. (2013). Real and Fictive Outcomes Are Processed Differently but Converge on a Common Adaptive Mechanism. *Neuron*, *79*(6), 1243–1255. <https://doi.org/10.1016/j.neuron.2013.07.006>
- Holroyd, C. B., Pakzad-Vaezi, K. L., & Krigolson, O. E. (2008). The feedback correct-related positivity: Sensitivity of the event-related brain potential to unexpected positive feedback. *Psychophysiology*, *45*(5), 688–697. <https://doi.org/10.1111/j.1469-8986.2008.00668.x>
- Kirschner, H., Fischer, A. G., & Ullsperger, M. (2022). Feedback-related EEG dynamics separately reflect decision parameters, biases, and future choices. *NeuroImage*, *259*, 119437. <https://doi.org/10.1016/j.neuroimage.2022.119437>
- Proudfit, G. H. (2015). The reward positivity: From basic research on reward to a biomarker for depression. *Psychophysiology*, *52*(4), 449–459. <https://doi.org/10.1111/psyp.12370>
- Schlagenhauf, F., Huys, Q. J. M., Deserno, L., Rapp, M. A., Beck, A., Heinze, H.-J., Dolan, R., & Heinz, A. (2014). Striatal dysfunction during reversal learning in unmedicated schizophrenia patients. *NeuroImage*, *89*, 171–180. <https://doi.org/10.1016/j.neuroimage.2013.11.034>
- Zhang, L., Lengersdorff, L., Mikus, N., Gläscher, J., & Lamm, C. (2020). Using reinforcement learning models in social neuroscience: frameworks, pitfalls and suggestions of best practices. *Social Cognitive and Affective Neuroscience*, *15*(6), 695–707. <https://doi.org/10.1093/scan/nsaa089>

Reviewer 1

1. **The authors' revisions have thoroughly addressed nearly all of my concerns about the previous version of the manuscript. I only have a few remaining comments.**
 - a. **The revised manuscript now reports that the reward positivity is evaluated in a 145-200 ms window post-feedback. I wonder why the authors chose this window? It is not the typical time period to observe the reward positivity, and in fact their own data suggest that the RewP peaks at 200 ms (see, e.g., Fig 3A). In fact, the reward positivity is normally seen around 240-340 ms, so why does it occur so early in this study? Is it because of the use of auditory feedback rather than visual feedback (resulting in faster latency)?**

The reviewer raises excellent points regarding the time window of our RewP waveform. In response, we reviewed the derivation of our waveforms and the associated analysis and found that we had simply misreported the windows in the main text. Our analysis was based on signals measured across the 165 – 225 ms post-feedback window in humans, and the 60 – 160 ms window in Rodents. These windows align with the difference wave peaks observed in Figures 2A & B of the main text, as well as the coefficient peaks shown in Figure 3. We have amended the manuscript to report the correct time windows. We are very grateful to the reviewer for their astute observation, which allowed us to rectify a typographical mistake in the first resubmission.

Regarding the difference in our RewP time window and the more typical 240-340 ms window (as suggested by the Reviewer and noted by Sambrook & Goslin (2015)), we believe that the reviewer is correct and prescribe this difference to the use of auditory feedback rather than the more typical visual feedback. As additional corroboration, using unpublished data from an independent cohort of humans (rodent data are also available) that completed the same PRL task, we find again find that RewP was earlier than the 240-340ms window and peaks around 190 – 200 ms in a close replication of the present waveforms (see Figure A overpage).

Figure A – Feedback-locked grand-average ERP waveforms recorded during the completion of the PRL task.

Note: Underlying EEG data are from an independent but comparable sample of $n = 25$ healthy adults who completed the same PRL as in the paper. Waveforms are plotted at electrode 2 in the montage which approximates the position of FCz. As can be seen, the expectancy-based RewPs peak around 190-200 ms post-feedback stimulus.

2. Fig 2A and 2B plot the target difference wave and the non-target difference wave, but these are not standard ways of assessing the reward positivity because they confound valence with expectancy. For example, the target difference wave compares a frequent reward with an infrequent no-reward, so the difference can be due to frequency rather than to reward. By contrast, Figure A in the authors' cover letter depicts the expected and unexpected difference waves, which are the more standard way of measuring the RewP because these comparisons remove the probability confound. I recommend that Fig 2A and 2B plot the expected and unexpected difference waves rather than the target and non-target difference waves. Parenthetically, the caption to Fig 2A and Fig 2B should describe the legend.

As recommended by the reviewer, we have amended Plots A and B of Figure 2 to present the expectancy-based difference waves that were included in our previous response; we have also amended Figure 2's caption. Furthermore, we have also made

the same amendments to Figure 4 and Supplementary Figure 6, which also previously presented reward-based difference waves.

Reviewer #2 :

1. The authors have successfully addressed all of my prior concerns.

We appreciate the reviewer's time and efforts in reviewing our manuscript.

2. One minor note: In Figure 2B, there is no "Target Difference" solid black line. Furthermore, the RewP label and arrow appears to be pointing to the red negative deflection, which is confusing.

We appreciate the reviewers' attention to detail. In response to comment 2 from Reviewer 1 we have amended Figure 2B to present expectancy-based difference waves rather than reward-based waves. In doing so, we paid particular attention to ensure that all relevant waves are plotted.

Reviewer #3:

1. The authors provide a substantially revised version of their manuscript in which they addressed all reviewers' concerns. I think the manuscript is now ready for publication. I have no further suggestions or comments.

We thank the reviewer for their time and efforts in reviewing our manuscript, as well as their recommendation.

References

Sambrook, T. D., & Goslin, Jeremy. (2015). A neural reward prediction error revealed by a meta-analysis of ERPs using great grand averages. *Psychological Bulletin*, 141(1), 213–235. <https://doi.org/10.1037/bul0000006>

Reviewer 1

- 1. The authors have thoroughly addressed my concerns. My only remaining suggestion is that the authors should comment on the early latency of the RewP in the discussion section. This is a rather peculiar finding that I suspect has more to do with the task design than with the use of auditory feedback (though it could be a combination of both). Because latency is a key signature for identifying a component, I believe it important that this discrepancy be acknowledged.**

We appreciate the reviewer's time, efforts, and expertise in reviewing our manuscript. As suggested, we have added a comment regarding the latency of our RewP and how it differs from the more canonical description to the discussion.